# A FAST AND EFFECTIVE ALTERNATIVE TO GRAPH TRANSFORMERS

## ABSTRACT

Graph Neural Networks (GNNs) have shown impressive performance in graph representation learning. However, GNNs face challenges in capturing long-range dependencies that limit their expressive power. To tackle this challenge, Graph Transformers (GTs) were introduced that utilize the self-attention mechanism to effectively model pairwise node relationships. Despite their advantages, GTs typically suffer from quadratic complexity with respect to the number of nodes in the graph, hindering their applicability to large graph datasets. In this work, we present Graph-Enhanced Contextual Operator (GECO), a fast and effective alternative to GTs that leverages shallow neighborhood propagation and global convolutions to effectively capture local and global dependencies. Evaluations on extensive collection of benchmarks showcase that GECO consistently achieves superior or comparable quality compared to the existing GTs across graphs of various types and scales, improving the SOTA up to $4.5\%$. Remarkably, these accomplishments are realized while maintaining quasilinear time and memory scaling, making GECO a promising solution for large-scale graph representation learning.

## 1 INTRODUCTION

Graph Neural Networks (GNNs) have been state-of-the-art (SOTA) models for graph representation learning showing superior performance across different tasks spanning node, link, and graph level prediction (Gori et al., 2005; Scarselli et al., 2009; Kipf & Welling, 2017; Zhang & Chen, 2018; Zhang et al., 2018). Despite their success, GNNs have fundamental limitations that affect their ability to capture long-range dependencies in graphs. These dependencies refer to nodes needing to exchange information over relatively long distances effectively, especially when the distribution of edges is not directly related to the task or when there are missing edges in the graph (Dwivedi et al., 2022b). This limitation can further lead to information over-squashing caused by repeated propagations within GNNs (Li et al., 2018; Alon & Yahav, 2020; Topping et al., 2022).

Graph Transformers (GTs) (Dwivedi & Bresson, 2020; Ying et al., 2021; Wu et al., 2021) were introduced to overcome the limitations of GNNs by incorporating the self-attention mechanism (Vaswani et al., 2017). GTs can model long-range dependencies by attending to potential neighbors among the entire set of nodes. They have shown remarkable performance, achieving the SOTA across various benchmarks. However, GTs also come with a notable drawback with their quadratic complexity compared to linear time and memory complexity inherent in GNNs. This quadratic complexity stems from that each node needs to attend to every other node, preventing GTs' widespread adoption in large-scale real-world scenarios. As mini-batch sampling methods for GTs still remain under-explored, the primary application of GTs has been on smaller datasets, such as molecular ones (Freitas et al., 2021; Hu et al., 2021; Dwivedi et al., 2022b; 2023). Consequently, exploring novel efficient and high-quality attention replacements remains a crucial research direction to unlock the full potential of GTs for large-scale graphs. Recently, global convolutional language models have emerged as promising alternatives for attention (Li et al., 2023). Specifically, Hyena (Poli et al., 2023) has demonstrated impressive performance, offering efficient processing of longer contexts with high quality, with a remarkable speed-up compared to optimized attention (Dao et al., 2022), and already motivated domain adaptations (Nguyen et al., 2023).

In this work, we aim to find efficient alternatives to dense attention mechanisms in order to scale graph transformers. Inspired by the efficiency and quality of the convolution based sequence models,

we develop a global convolutional model encompassing all nodes to replace dense attention in graph domain. The challenge is to design a scalable global convolution model that can effectively capture both local and long-range dependencies in large graphs. To this end, we introduce Graph-Enhanced Contextual Operator (GECO), a scalable graph attention replacement consisting of four main components: (1) local propagation to capture local context, (2) global convolution to capture global context with quasilinear complexity, (3) data-controlled gating for context-specific operations on each node, (4) positional/structural encoder for feature encoding and graph ordering. Our evaluation focuses on two main objectives: surpassing SOTA GT quality on smaller graph datasets emphasized by the community, and scaling to larger graph datasets where traditional attention mechanisms are impractical to apply. We have conducted experiments on various benchmark datasets with various graph tasks and scales, and results show that GECO consistently delivers strong performance and often achieves either SOTA or competitive quality. The main contributions of this paper include:

- To our best knowledge, our work has been the first to develop a scalable and effective alternative to self-attention based GTs that boosts computation efficiency while maintaining prediction quality, and in most cases, leading to improvements.

- We developed GECO as a compact layer consisting of local propagation and global convolution model blocks with quasilinear time complexity. Unlike previous methods, GECO is a refined layer without intermediate parameters and non-linearities between local and global blocks, applying skip connections to the layer as a whole.

- We demonstrate GECO scales to large-scale graphs that are infeasible for existing GTs utilizing self-attention due to their intrinsic quadratic complexity, and further improves prediction accuracy by up to 4.5% for large graph datasets.

- We demonstrated GECO is capable of capturing long-range dependencies in graphs. In our experiments, GECO has achieved SOTA on a majority of long-range benchmark datasets, improving baselines by up to $4.3\%$.

## 2 BACKGROUND AND RELATED WORK

### 2.1 GRAPH NEURAL NETWORKS (GNNS)

A graph $G = (V, E)$, comprises a set of vertices $V$ and edges $E \subseteq V \times V$. $A \in \mathbb{R}^{N \times N}$ is the adjacency matrix and a weighted edge $(u \rightarrow v) \subseteq E$ exists between source $u$ and target $v$ vertices if $A_{u,v} \neq 0$. The node feature matrix $X^{(0)} \in \mathbb{R}^{N \times d^{(0)}}$ maps $v$ to a feature vector $x_v^{(0)} \in \mathbb{R}^{d^{(0)}}$. $\mathcal{N}(v) = \{u \mid (u \rightarrow v) \in E\}$ is the incoming neighbors of $v$. GNNs adopt an AGGREGATE-COMBINE framework (Hamilton et al., 2017) to compute layer-$l$ representation $h_v^{(l)}$ such that:

$$h_v^{(l)} = \text{COMBINE}^{(l)}\big(\alpha_v^{(l)}, h_v^{(l-1)}\big), \qquad \alpha_v^{(l)} = \text{AGGREGATE}^{(l)}\big(\{h_u^{(l-1)} : u \in \mathcal{N}(v)\}\big) \quad (1)$$

Additionally, a pooling function generates graph representation, $h_G = \text{POOL}\big(\{h_v^{(L)} | v \in V\}\big)$.

**Challanges.** GNNs efficiently scale to large graphs with linear complexity of $\mathcal{O}(|V| + |E|)$. However, they face challenges capturing long-range dependencies, often requiring many hops and nonlinearities for information to traverse distant nodes, depending on the graph's topology (Alon & Yahav, 2020; Dwivedi et al., 2022b). GTs effectively resolve this issue through dense pairwise attention.

### 2.2 GRAPH TRANSFORMERS (GTS)

**Multi-head Attention (MHA).** At the core of Transformer lies the multi-head self-attention (MHA) (Vaswani et al., 2017), which maps the input $H \in \mathbb{R}^{N \times d}$ to $\mathbb{R}^{N \times d}$ as follows:

$$Attn(H) = \text{Softmax}\left(\frac{QK^T}{\sqrt{d}}\right), \qquad y = \text{SelfAttention}(H) = Attn(H)V \quad (2)$$

where *query* $(Q = HW_q)$, *key* $(K = HW_k)$, and *value* $(V = HW_v)$ are projections of the input, $W_q, W_k, W_v \in \mathbb{R}^{d \times d}$. The attention matrix $Attn(H)$ captures the pair-wise similarities of the input.

**Graph Transformers (GTs)** generalize Transformers to graphs. Depending on their *focus of attention*, we group GTs into three categories. **Sparse GTs** employ the adjacency matrix as an attention mask, allowing nodes to pay attention to *their neighbors*, which facilitates weighted neighborhood aggregation (Veličković et al., 2018; Xu et al., 2019). **Layer GTs** use a GNN to generate hop-tokens for nodes, followed by MHA on these tokens, where nodes pay attention to *their layer embeddings*, resembling specific instances of JK-Nets (Xu et al., 2018), as explained in Appendix E.4. NAG-phormer (Chen et al., 2023) uses an offline GNN (feature propagation). While both Sparse and Layer GTs utilize attention, they still face long-range dependency problem as their attention is restricted to nodes within a fixed number of hops. Comparatively, **Dense GTs** perform attention on fully connected graphs, enabling nodes to pay attention to *all the other nodes* regardless of their distances. GT (Dwivedi & Bresson, 2020) incorporates Laplacian eigen-vectors (LE) as positional encodings (PE) to enrich node features with graph information. SAN (Kreuzer et al., 2021) introduces learnable LE through permutation-invariant aggregation. Graphormer (Ying et al., 2021) uses node degrees as PE and shortest path distances as relative PE. GraphiT (Mialon et al., 2021) proposes relative PE based on diffusion kernels. SAT (Chen et al., 2022) extracts substructures centered at nodes as additional tokens, while (Zhao et al., 2023) uses substructure-based local attention with substructure tokens. GOAT (Kong et al., 2023) uses dimensionality reduction to reduce the computational cost of attention. (Diao & Loynd, 2023) enhances attention with additional edge updates.

**Challenges.** Dense GTs introduce computational and memory bottlenecks due to their increased complexity from $\mathcal{O}(N + M)$ to $\mathcal{O}(N^2)$ where $M = |E|$, restricting their application to large graphs. GraphGPS (Rampasek et al., 2022) offers a modular framework that combines GNNs with a global attention module, including subquadratic transformer approximations (Zaheer et al., 2020; Kreuzer et al., 2021). Unfortunately, these subquadratic models show lower quality, and MHA-based quadratic methods struggle with scalability. Consequently, finding a suitable subquadratic attention mechanism replacement remains a challenge, and this work is dedicated to addressing this problem.

**Other Related Work.** Exphormer (Shirzad et al., 2023) enhances GraphGPS by utilizing GNNs on original graphs and Transformers on expander graphs. (He et al., 2023) generalizes ViT (Dosovitskiy et al., 2020) and MLP-Mixer (Tolstikhin et al., 2021) to graphs. (Zhang et al., 2022) proposes a node sampling strategy formulated as an adversary bandit problem. HSGT (Zhu et al., 2023) proposes a coarsening strategy to scale to large-scale graphs, effectively learning multi-level hierarchies.

## 2.3 Attention Alternatives

A circular convolution of input $u_t \in \mathbb{R}^N$ and a filter $z \in \mathbb{R}^N$ is expressed by:

$$y_t = (u * z)_t = \sum_{i=0}^{N-1} u_i z_{(t-i) \bmod N} \tag{3}$$

here we assume that both input and output have a single channel. Convolutions are crucial in deep learning with the primary approach being the optimization of convolution filters $z_t$. Convolutional Neural Networks (LeCun et al., 1998) optimize $z_t$ at every $K$ steps, where $K$ denotes the filter size. This *explicit* parametrization captures local patterns within every $K$ steps. Alternatively, *implicit* parameterizations represent the filter as a learnable function such that $z_t = \gamma_\theta(t)$. Diverse parametrization techniques have emerged, advancing the SOTA on various benchmarks (Tay et al., 2021; Gu et al., 2021; Romero et al., 2022; Smith et al., 2023; Fu et al., 2023b;a). Recently, Hyena (Poli et al., 2023), an implicitly parametrized model utilizing global convolutions and gating, stands out with its quality matching Transformer in quasilinear time.

Equation 3 can be also expressed as $y_t = Tu$, where $T \in \mathbb{R}^{N \times N}$ is Toeplitz kernel matrix induced by filter $z$. Then the second-order Hyena operator with input $x \in \mathbb{R}^{N \times 1}$ defined as follows:

$$y = \text{Hyena}(x_1, x_2)x_3 \quad and \quad \text{Hyena}(x_1, x_2) = D_{x_2} T D_{x_1} \tag{4}$$

where $x_1$, $x_2$, and $x_3$ are all projections of the input $x$, and $T \in \mathbb{R}^{N \times N}$ is used as a learnable convolution filter. In this context, $T$ is learned by a neural network, where $T_{uv} = z_{u-v} = z_t = \gamma_\theta(t)$. $D_{x_1}, D_{x_2} \in \mathbb{R}^{N \times N}$ are diagonal matrices with $x_1$ and $x_2$ on their diagonals respectively.

**Connection to Attention.** $\text{Hyena}(x_1, x_2)$ acts similar to attention matrix at Equation 2, however, it is realized by interleaving global convolutions and element-wise gating. Furthermore, $y = \text{Hyena}(x_1, x_2)x_3$ is efficiently computed without materializing the full matrix, using two FFT convolutions and two gating. Please refer to (Poli et al., 2023) for more details.

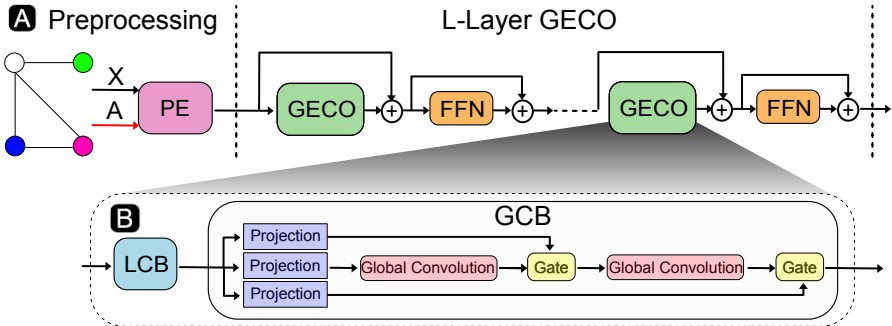

Figure 1: **A** Our architecture comprises Positional Encoding (PE) block and multiple layers of Graph-Enhanced Contextual Operators (GECOs): PE block extracts positional/structural encodings and updates node features as a preprocessing step, and each GECO layer is followed by a FFN. **B** A GECO unit contains a Local Propagation Block (LCB) that aggregates neighborhood embeddings and concatenates with original ones to capture local dependencies, as well as a Global Context Block (GCB) that efficiently captures global dependencies using graph aware global convolutions.

## 3 GECO: A FAST AND EFFECTIVE ALTERNATIVE TO GTS

We present Graph-Enhanced Contextual Operator (GECO), a novel paradigm to replace dense attention in quasilinear time and memory complexity. It draws inspiration from recent advancements in global convolution models, and offers a promising approach to capture long-range dependencies with subquadratic operators. As illustrated in Figure 1, GECO starts with positional/structural encodings, and proceeds through multiple layers of GECO, followed by a point-wise feed-forward neural network (FFN). We introduce the main components in the following subsections.

### 3.1 STRUCTURAL/POSITIONAL ENCODINGS

Structural and positional encodings play a pivotal role in the realm of graph transformers. In our approach, we follow the foundational work established in prior literature concerning these encodings (Dwivedi & Bresson, 2020; Kreuzer et al., 2021; Ying et al., 2021; Dwivedi et al., 2022a). To seamlessly integrate these encodings with the original input features, we employ a concatenation method. Given a positional/structural encoding matrix $U \in \mathrm{R}^{N \times d_u}$, where $d_u$ represents the encoding dimension, we combine it with the original node features denoted by $X$. This concatenation process results in a new feature matrix $X^*$, defined as follows: $X^* = [X, U]$. For further details on incorporating relative encodings, please refer to Appendix B.1.

### 3.2 LOCAL PROPAGATION BLOCK (LCB)

Local Propagation Block (LCB) aggregates neighborhood embeddings for each node and concatenates them with the original ones. Notably, *no parameters* are involved at this stage, making it akin to the traditional feature propagation with a dense skip connection. LCB can be expressed:

$$h_v^{*(l)} = [\alpha_v^{(l)}, h_v^{(l-1)}] \quad or \quad H^{*(l)} = [H^{(l)}, AH^{(l)}] \tag{5}$$

where $\alpha_v^{(l)}$ and $h_v^{(l-1)}$ are defined as before. Instead of adding self-edges for each node, we concatenate $h_v^{(l)}$ and $\alpha_v^{(l)}$, enabling our model to distinguish node and propagation embeddings. By leveraging the LCB, nodes are prepared for evaluation in local and global contexts. Moreover, rather than solely relying on $h_v^{(l)}$, local attention mechanisms similar to those found in GAT (Veličković et al., 2018) can be incorporated. Alternative approaches for LCB are further discussed in Section 4.3.

### 3.3 GLOBAL CONTEXT BLOCK (GCB)

Efforts have been directed at creating efficient attention alternatives to capture longer contexts through techniques like low-rank approximation, factorization and sparsification, often leading to

trade-offs between efficiency and quality (Catania et al., 2023). Meanwhile, recent sequence models opt for linear convolutions or RNNs which offers near-linear time complexity (Gu et al., 2021; Fu et al., 2023a; Peng et al., 2023; Poli et al., 2023). Building upon the evolving research, for the first time, we explore whether global convolutions can capture global context within graph structures. To address this, we developed a modified graph-aware global convolution operator based on Hyena.

Standard Hyena is designed for sequences and lacks graph handling capabilities inherently. This leads us to a key question: Can we create an operator for efficient graph processing using global convolutions? Our investigation yielded positive results, leading to Global Context Block (GCB), a novel operator with graph awareness. We outline its key distinctions and enhancements compared to the original Hyena below:

- *Graph-to-sequence conversion:* Given our focus on graphs, we arrange both $A$ and $X$ using permutation $P$ and convert them into time-correlated sequences, aligning node IDs with time $(t)$.
- *Bidirectional information flow:* Since our setup lacks causality unlike language modeling, we remove the causal mask from the Hyena operator. This allows information to flow bidirectionally and respects the natural dynamics of graph data.
- *Graph-aware context:* (1) We remove Hyena's short convolution along the sequence length, as permutation $P$ might not reflect a locality-sensitive order. Instead, GECO uses LCB for local and GCB for global dependency modeling. (2) The original algorithm uses exponential decay modulation for global convolution filters, giving higher weights to nearby points in the sequence. In contrast, we omit this decay and treat all nodes equally, regardless of their distance under permutation P. This choice stems from our use of node IDs as positional encodings in our setup.
- *Window of the global convolution:* We set the window size for global convolutions to match the number of nodes in the graph, ensuring the inclusion of all nodes within the convolution operation. This is similarly reasoned by the natural dynamics of graph data where node permutations do not introduce proximity-based context.

---

**Algorithm 1** Forward pass of GCB Operator

---

**Input:** Node embeddings $\mathbf{X} \in \mathbb{R}^{N \times d}$; Order $K$; PE dim $d_e$;
  1. $P_1, \ldots, P_K, V = \mathsf{Projection}(\mathbf{X})$ *# Linear projection function, $P_i$ for projections*
  2. $F_1, \ldots, F_K = \mathsf{Filter}(N, d_e)$ *# Position dependent filter function, $F_i$ for filters*
  *# Update Value, $V$, until all projections are exhausted*
  **for** $i = 1, \ldots, K$ **do**
    3. In parallel across $d$: $V_t \leftarrow (P_i)_t \cdot \mathsf{FFTConv}(F_i, V)_t$ *# Call to FFT-Conv, t is the position of convolution*
  **end for**
  4. Return $V$

---

Algorithm 1 presents the GCB algorithm, with adapted and modified from the original work (Poli et al., 2023), with unified notation. For a given node embedding matrix $X$, GCB generates $(K + 1)$ projections, where $K$ is a hyperparameter controlling the operator's recurrence. Throughout this work, we set $K = 2$, and in this specific instance, the three projections serve roles similar to query, key, and value. For each projection, a filter is learned by a simple FFN, with node IDs utilized for filters' positional encoding. Subsequently, we update the value $V$ using global convolutions with one projection and filter at a time, followed by element-wise gating, until all projections are processed.

### 3.4 END-TO-END TRAINING

Algorithm 2 presents the end-to-end training process. We start by encoding positional information and permuting the graph using the given permutation as a preprocessing step. The training procedure of GECO can be broken down into two main blocks, as described above. LCB begins by propagating neighborhood embeddings and applying normalization and followed by GCB. Following each GECO block, a standard FFN block is employed, which consists of two dense linear layers with an activation function and dropout applied between the layers, such that:

$$\mathrm{FFN}(X) = \sigma(XW_1)W_2 \tag{6}$$

$W_1, W_2 \in \mathbb{R}^{d \times d}$ represent the weights for projections. Both the GECO and FFN implement residual connections, normalization, and dropout. GECO uses three quasilinear operators and can be

---

**Algorithm 2** End-to-end `GECO` Model Training

---

**Input:** Adjacency matrix $\mathbf{A} \in \mathbb{R}^{N \times N}$; Node features $\mathbf{X} \in \mathbb{R}^{N \times d}$; Permutation $\mathbf{P} \in \mathbb{R}^{N \times N}$; Edge features $\mathbb{E} \in \mathbb{R}^{M \times d_e}$.

1. $\mathbf{X}, \mathbf{A} = GraphPositionalEncoder(\mathbf{X}, \mathbf{A}, \mathbb{E})$
2. $\mathbf{X}^{(0)} = Permute(\mathbf{A}, \mathbf{X}, \mathbf{P})$

**for** $\ell = 0, \dots, L-1$ **do**

    3. $\mathbf{X}^{(l+1)} = LayerNorm(GECO(\mathbf{X}^{(l)}, \mathbf{A}) + \mathbf{X}^{(l)})$

    4. $\mathbf{X}^{(l+1)} = LayerNorm(FFN(\mathbf{X}^{(l+1)}) + \mathbf{X}^{(l+1)})$

**end for**

5. Return $\mathbf{X}^L \in \mathbb{R}^{N \times D}$

---

computed in $\mathcal{O}(N \log N + M)$. For the complete algorithm and complexity analysis, please refer to Appendices B.2 and C.2 respectively.

### 3.5 PITFALLS OF PERMUTATION SENSITIVITY

The GECO presents certain pitfalls that need to be considered in terms of permutation sensitivity. While typical GNNs are implemented using permutation invariant functions (Kipf & Welling, 2017; Hamilton et al., 2017), both short and global convolutions of the GCB operator are shift invariant but not permutation invariant. By replacing short convolutions with propagation, we make the local mixing permutation invariant. However, the global convolutions remain order sensitive. To address this challenge, we have experimented with different orderings, including random permutations. Surprisingly, we observed that the final results are not significantly impacted by different orderings. While we observe reasonable robustness to different orderings, understanding and addressing this limitation is crucial for broadening the applicability of GECO.

Importantly, a line of research focuses on order-sensitive GNNs (Chen et al., 2020; Huang et al., 2022; Chatzianastasis et al., 2023) to achieve greater model expressibility. Remarkably, Graph-SAGE (Hamilton et al., 2017) with LSTM aggregator is also order-sensitive and has shown impressive results across multiple datasets. However, while order-sensitive approaches may improve performance for a specific task, the model could potentially lose its generalizability.

### 3.6 COMPARISON WITH HYBRID APPROACHES

Previous approaches consisting of local and global modules (Wu et al., 2021; Dwivedi & Bresson, 2020; Lin et al., 2021; Min et al., 2022) combine off-the-shelf GNN and Transformer arhictectures straightforwardly. In contrast, GECO's LCB and GCB are not auxiliary modules. GECO designed as a new compact layer comprising local and global blocks, simplifying the model by removing intermediate parameters and non-linearities. Consequently, it uses skip connections to the entire layer as a whole, rather than separately. Please refer to Appendices E.2 and E.3 for further details.

## 4 EXPERIMENTS

### 4.1 PREDICTION QUALITY

We assess the GECO on ten benchmarks outlined in Appendix A.1, where each graph dataset contains multiple graphs and each graph has an average number of nodes ranging from tens to five hundred. We start by creating hybrid GNN+GECO replacing the attention module used in GraphGPS. For dataset and hyperparameter details refer to Appendix A and Appendix D.3, respectively.

**Long Range Graph Benchmark (LRGB).** Table 1 outlines our evaluation on the LRGB, which comprises graph tasks designed to assess a model's capacity to capture long-range dependencies. The results indicate that GECO outperforms baselines across the majority of the datasets by upto $4.3\%$. For the remaining datasets, it consistently ranks within the top three, with performance within $1.3\%$ of the best baseline. In summary, GECO effectively captures long-range dependencies and replaces MHA without compromising quality, often surpassing it. Interestingly, GECO's F1 score on PascalVOC increased from $0.4053$ to $0.4210$ when positional encodings are discarded, resulting in enhanced quality and a simplified model.

Table 1: LRGB Evaluation: the **first**, **second**, and **third** best are highlighted. We reuse the results reported at (Rampasek et al., 2022) except for Exphormer (Shirzad et al., 2023).

| Model | PascalVOC-SP | COCO-SP | Peptides-func | Peptides-struct | PCQM-Contact |
|---|---|---|---|---|---|
| | F1 score ↑ | F1 score ↑ | AP ↑ | MAE ↓ | MRR ↑ |
| GCN | $0.1268 \pm 0.0060$ | $0.0841 \pm 0.0010$ | $0.5930 \pm 0.0023$ | $0.3496 \pm 0.0013$ | $0.3234 \pm 0.0006$ |
| GINE | $0.1265 \pm 0.0076$ | $0.1339 \pm 0.0044$ | $0.5498 \pm 0.0079$ | $0.3547 \pm 0.0045$ | $0.3180 \pm 0.0027$ |
| GatedGCN | $0.2873 \pm 0.0219$ | $0.2641 \pm 0.0045$ | $0.5864 \pm 0.0077$ | $0.3420 \pm 0.0013$ | $0.3218 \pm 0.0011$ |
| GatedGCN+RWSE | $0.2860 \pm 0.0085$ | $0.2574 \pm 0.0034$ | $0.6069 \pm 0.0035$ | $0.3357 \pm 0.0006$ | $0.3242 \pm 0.0008$ |
| Transformer+LapPE | $0.2694 \pm 0.0098$ | $0.2618 \pm 0.0031$ | $0.6326 \pm 0.0126$ | $0.2529 \pm 0.0016$ | $0.3174 \pm 0.0020$ |
| SAN+LapPE | $0.3230 \pm 0.0039$ | $0.2592 \pm 0.0158$ | $0.6384 \pm 0.0121$ | $0.2683 \pm 0.0043$ | $0.3350 \pm 0.0003$ |
| SAN+RWSE | $0.3216 \pm 0.0027$ | $0.2434 \pm 0.0156$ | $0.6439 \pm 0.0075$ | $0.2545 \pm 0.0012$ | $0.3341 \pm 0.0006$ |
| GPS | $0.3748 \pm 0.0109$ | $0.3412 \pm 0.0044$ | $0.6535 \pm 0.0041$ | $0.2500 \pm 0.0005$ | $0.3337 \pm 0.0006$ |
| Exphormer | $0.3975 \pm 0.0037$ | $0.3455 \pm 0.0009$ | $0.6527 \pm 0.0043$ | $0.2481 \pm 0.0007$ | $0.3637 \pm 0.0020$ |
| GECO (Ours) | $0.4210 \pm 0.0080$ | $0.3320 \pm 0.0032$ | $0.6975 \pm 0.0025$ | $0.2464 \pm 0.0009$ | $0.3526 \pm 0.0016$ |

Table 2: OGBG Evaluation: the **first**, **second**, and **third** best are highlighted. We reuse results reported by (Rampasek et al., 2022).

| Model | ogbg-molhiv | ogbg-molpcba | ogbg-ppa | ogbg-code2 |
|---|---|---|---|---|
| | AUROC ↑ | Avg. Precision ↑ | Accuracy ↑ | F1 score ↑ |
| GCN+virtual node | $0.7599 \pm 0.0119$ | $0.2424 \pm 0.0034$ | $0.6857 \pm 0.0061$ | $0.1595 \pm 0.0018$ |
| GIN+virtual node | $0.7707 \pm 0.0149$ | $0.2703 \pm 0.0023$ | $0.7037 \pm 0.0107$ | $0.1581 \pm 0.0026$ |
| GatedGCN-LSPE | – | $0.2670 \pm 0.0020$ | – | – |
| PNA | $0.7905 \pm 0.0132$ | $0.2838 \pm 0.0035$ | – | $0.1570 \pm 0.0032$ |
| DeeperGCN | $0.7858 \pm 0.0117$ | $0.2781 \pm 0.0038$ | $0.7712 \pm 0.0071$ | – |
| DGN | $0.7970 \pm 0.0097$ | $0.2885 \pm 0.0030$ | – | – |
| GSN (directional) | $0.8039 \pm 0.0090$ | – | – | – |
| GSN (GIN+VN base) | $0.7799 \pm 0.0100$ | – | – | – |
| CIN | $0.8094 \pm 0.0057$ | – | – | – |
| GIN-AK+ | $0.7961 \pm 0.0119$ | $0.2930 \pm 0.0044$ | – | – |
| CRaWl | – | $0.2986 \pm 0.0025$ | – | – |
| ExpC | $0.7799 \pm 0.0082$ | $0.2342 \pm 0.0029$ | $0.7976 \pm 0.0072$ | – |
| SAN | $0.7785 \pm 0.2470$ | $0.2765 \pm 0.0042$ | – | – |
| GraphTrans (GCN-Virtual) | – | $0.2761 \pm 0.0029$ | – | $0.1830 \pm 0.0024$ |
| K-Subtree SAT | – | – | $0.7522 \pm 0.0056$ | $0.1937 \pm 0.0028$ |
| GPS | $0.7880 \pm 0.0101$ | $0.2907 \pm 0.0028$ | $0.8015 \pm 0.0033$ | $0.1894 \pm 0.0024$ |
| GECO | $0.7980 \pm 0.0200$ | $0.2961 \pm 0.0008$ | $0.7982 \pm 0.0042$ | $0.1915 \pm 0.002$ |

**Open Graph Benchmark (OGB).** Table 2 presents the evaluation results for our GECO on four OGB Graph (OGBG) prediction datasets, encompassing both molecular and code graphs. Similar to GraphGPS, we observed instances of overfitting within the GECO. Nevertheless, GECO outperforms GraphGPS on the majority of the datasets, except for ppa. Across all datasets, it consistently secures the top three, demonstrating its effectiveness as a high-quality and efficient GT alternative.

Table 3: PCQM4Mv2 evaluation: the **first**, **second**, and **third** best are highlighted. *Validation* set is used for evaluation as *test* labels are private. We reuse results reported by (Rampasek et al., 2022).

| Model | PCQM4Mv2 | | | |
|---|---|---|---|---|
| | Test-dev MAE ↓ | Validation MAE ↓ | Training MAE | # Param. |
| GCN | 0.1398 | 0.1379 | n/a | 2.0M |
| GCN-virtual | 0.1152 | 0.1153 | n/a | 4.9M |
| GIN | 0.1218 | 0.1195 | n/a | 3.8M |
| GIN-virtual | 0.1084 | 0.1083 | n/a | 6.7M |
| GRPE | 0.0898 | 0.0890 | n/a | 46.2M |
| EGT | 0.0872 | 0.0869 | n/a | 89.3M |
| Graphormer | n/a | 0.0864 | 0.0348 | 48.3M |
| GPS-small | n/a | 0.0938 | 0.0653 | 6.2M |
| GPS-medium | n/a | 0.0858 | 0.0726 | 19.4M |
| GECO | n/a | 0.08413 | 0.05782 | 6.2M |

**PCQM4Mv2.** Table 3 outlines the evaluation on PCQM4Mv2. Our findings show that GECO significantly enhances the performance of both GNN and GT baselines. Remarkably, GECO achieves

Table 4: Accuracy on large node prediction datasets: the **first**, **second**, and **third** best are highlighted. We reuse the results reported by (Han et al., 2023; Shirzad et al., 2023; Zeng et al., 2021).

| Model Accuracy | Flickr Accuracy | Arxiv Accuracy | Reddit Accuracy | Yelp Micro-F1 Score |
|---|---|---|---|---|
| GCN | $50.90 \pm 0.12$ | $70.25 \pm 0.22$ | $92.78 \pm 0.11$ | $40.08 \pm 0.15$ |
| SAGE | $53.72 \pm 0.16$ | $72.00 \pm 0.16$ | $96.50 \pm 0.03$ | $63.03 \pm 0.59$ |
| GraphSaint | $51.37 \pm 0.21$ | $67.95 \pm 0.24$ | $95.58 \pm 0.07$ | $29.42 \pm 1.32$ |
| Cluster-GCN | $49.95 \pm 0.15$ | $68.00 \pm 0.59$ | $95.70 \pm 0.06$ | $56.39 \pm 0.64$ |
| GAT | $50.70 \pm 0.32$ | $71.59 \pm 0.38$ | – | – |
| GATv2 | – | $71.87 \pm 0.25$ | – | – |
| JK-Net | – | $72.19 \pm 0.21$ | – | – |
| Graphormer | OOM | OOM | – | OOM |
| Graphormer-SAMPLE | $51.93 \pm 0.21$ | $70.43 \pm 0.20$ | – | $60.01 \pm 0.45$ |
| SAN | OOM | OOM | – | OOM |
| SAT | OOM | OOM | – | OOM |
| SAT-SAMPLE | $50.48 \pm 0.34$ | $68.20 \pm 0.46$ | – | $60.32 \pm 0.65$ |
| ANS-GT | – | $68.20 \pm 0.46$ | – | – |
| GraphGPS | OOM | OOM | OOM | OOM |
| HSGT | $54.12 \pm 0.51$ | $72.58 \pm 0.31$ | – | $63.47 \pm 0.45$ |
| Exphormer | - | $72.44 \pm 0.28$ | – | – |
| GECO (Ours) | $55.55 \pm 0.25$ | $73.10 \pm 0.24$ | $96.65 \pm 0.05$ | $63.18 \pm 0.59$ |

this while using only **1/8** and **1/3** of the parameters compared to Graphormer and GraphGPS respectively. This parameter reduction brings GECO in close proximity to the parameter count used by GNN baselines, while substantially enhancing their performance.

## 4.2 SCALABILITY FOR LARGER GRAPHS

After achieving our initial objective of enhancing prediction quality on the above relatively small datasets, we assess GECO on 4 benchmark datasets where each graph contains a much larger number of nodes. Notably, traditional Dense GTs struggle to handle such large graphs due to their quadratic complexity while GECO succeeds with its superior computational and memory efficiency. In the following experiments, we design our models using only GECO blocks, following Algorithm 2. For simplicity, we avoid using structural/positional encodings as computing them may be infeasible for large graphs. For details on datasets and hyperparameters, please refer to Appendices A.2 and D.3.

Unlike previous works that exhibit a trade-off between quality and scalability, GECO efficiently scales and achieves superior quality across all datasets compared to Dense GTs (Graphormer/GraphGPS) that incur in OOM/timeout issues. Remarkably, GECO demonstrates significant predictive superiority, surpassing Dense GT baseline methods by up to 4.5%. On Arxiv, GECO outperforms recently proposed GT works Exphormer and GOAT (Kong et al., 2023) up to 0.7%. Notably, Graphormer with sampling falls short in achieving competitive quality across all datasets. Additionally, when comparing GECO to various baselines, including orthogonal methods, GECO remains competitive. It outperforms various baselines on Flickr, Arxiv, and Reddit, except for Yelp where the coarsening approach HSGT (Zhu et al., 2023) surpasses GECO. We leave the exploration of combining GECO with orthogonal methods such as expander graphs (Shirzad et al., 2023), hierarchical learning (Zhu et al., 2023), and dimensionality reduction (Kong et al., 2023) as future work to potentially get even better results. In summary, our results highlight that the global context can enhance the modeling quality for large node prediction datasets, justifying our motivation to find efficient high-quality attention alternatives. To the best of our knowledge, GECO is the first work attempting to capture pairwise node relations without heuristics at scale. Our experiments illustrate its effectiveness as a replacement for dense attention for large graphs.

## 4.3 ABLATION STUDIES

**Local Propagation Block Alternatives.** In GECO, we adopted LCB for graph-aware local context modeling instead of using 1D convolutions originally used in Hyena. This is motivated by the limitation of 1D convolutions in capturing local dependencies in graphs where node order does not imply proximity At Table 5, we focused on exploring alternatives to LCB within our GECO module. We experimented with replacing LCB with 1D convolutions of various filter sizes to help understand

Table 5: Ablation study on the LCB alternatives: the **first**, **second**, and **third** best are highlighted. Conv-$x$ indicates 1D Convolution with a filter size of $x$.

| Model | PascalVOC-SP | Peptides-func | Peptides-struct |
|---|---|---|---|
| | F1 score ↑ | AP ↑ | MAE ↓ |
| Transformer | **0.2762** | 0.6333 | **0.2525** |
| Performer | 0.2690 | 0.5881 | 0.2739 |
| GECO (Conv-1) | **0.2752** | 0.6589 | 0.2587 |
| GECO (Conv-10) | 0.1757 | **0.6819** | **0.2516** |
| GECO (Conv-20) | 0.1645 | **0.6706** | 0.2534 |
| GECO (Conv-40) | 0.1445 | 0.6517 | 0.2547 |
| GECO (LCB) | **0.3220** | **0.6876** | **0.2454** |

Table 6: Accuracy across large-scale datasets with different permutation strategies (Natural/Random) with GECO, alongside a comparison with default Hyena (Poli et al., 2023).

| Dataset | Hyena | GECO (Natural) | GECO (Random) |
|---|---|---|---|
| Flickr | $46.97 \pm 0.08$ | $55.55 \pm 0.25$ | $55.73 \pm 0.27$ |
| Arxiv | $56.04 \pm 0.61$ | $73.10 \pm 0.24$ | $73.08 \pm 0.28$ |
| Reddit | $69.24 \pm 0.54$ | $96.65 \pm 0.05$ | $96.62 \pm 0.05$ |
| Yelp | $50.08 \pm 0.31$ | $63.18 \pm 0.59$ | $63.23 \pm 0.50$ |

its effectiveness. We consistently observed a diminishing trend in quality as filter sizes increased, which can be attributed to larger filter sizes leading to a mix of unrelated nodes within the graph. In contrast, GECO with LCB consistently outperformed its alternatives as well as the Transformer and Performer, which highlights its effectiveness in capturing local graph dependencies.

**Permutation Robustness.** Next, we investigate the influence of node ordering permutations on GECO's performance. In our ablation study at Table 6, we first maintained the natural ordering of the graph and reported the mean and standard deviation of 10 runs with distinct seeds on this fixed permutation. Then, we repeated the same process, but we applied a random permutation to the graph once before training for each seed. Our results consistently demonstrate negligible differences in prediction accuracy between the two settings. While this study verifies GECO's robustness to various permutations, we acknowledge that adding a subquadratic operator that is invariant to permutations would be a valuable enhancement to our framework.

**Hyena Comparison.** In Table 6 compares GECO with off-the-shelf Hyena, by seting its filter size as the entire graph. The results demonstrate that GECO consistently outperforms off-the-shelf Hyena, which experiences significant quality declines. This highlights GECO's effectiveness in adapting global convolutions for large scale graph learning.

**Scaling Study.** Fig. 2 shows GECO's speedup w.r.t. the optimized attention, FlashAttention (Dao et al., 2022), for increasing numbers of nodes using synthetic datasets with similar sparsity patterns to those in Table 4. The results highlight that the speedup linearly increases with the number of nodes, and GECO reaches $169\times$ speedup on a graph with 2M nodes, confirming its relative scalability. Details including runtime numbers can be found in Appendix E.1.

## 5 CONCLUSION

We present GECO, a novel graph learning model that replaces the compute-intensive MHA in GTs with an efficient and high-quality operator. With a comprehensive evaluation, we demonstrate GECO: (1) replaces MHA without sacrificing quality and even outperforms the baselines on multiple graph benchmark datasets, (2) scales effectively to handle large datasets, and (3) enhances the quality of graph learning models by capturing global context while maintaining computational efficiency. Moving forward, we plan to investigate lightweight positional and structural encoding techniques, along with exploring alternative approaches for GCB, as well as model and data parallelism methods. These efforts aim to extend the scalability of our method to large graphs in the industry.

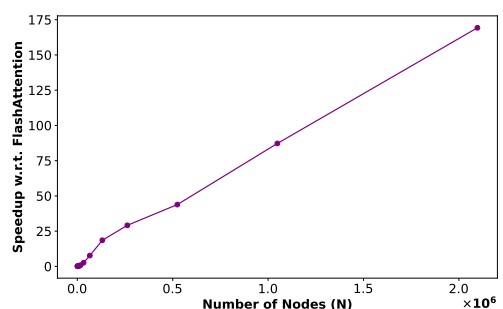

Figure 2: Relative speedup of GECO w.r.t. FlashAttention (Dao et al., 2022) on synthetic datasets. $\mathcal{O}(N^2/(N \log N)) = \mathcal{O}(N/\log N)$ characterizes the speedup.

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

# A  DATASETS

We gather a wide-ranging selection of 14 datasets, encompassing diverse graph types, tasks, and scales, collected from different sources. To facilitate understanding, we classify these datasets into two overarching groups. Each dataset in the first group comprises multiple graphs, each having small number of nodes/edges. On the other hand, the second group comprises node prediction datasets, each containing a single graph with much larger number of nodes.

## A.1  DATASETS WITH MULTIPLE GRAPHS

Table 7: Statics for the Datasets with Multiple Graphs, sorted by #average nodes.
**MRR**: Mean Reciprocal Rank, **AP**: Average Precision, **MAE**: Mean Absolute Error.
Sparsity is calculated as $\frac{M}{N^2}$, where N and M represent the average number of nodes and edges, respectively.

| Dataset | # Graphs | # Avg. Nodes | # Avg. Edges | Sparsity | Level | Task | Metric |
|---|---|---|---|---|---|---|---|
| **Long Range Graph Benchmark** | | | | | | | |
| PCQM-Contact | $529,434$ | $30.1$ | $61.0$ | $6.79 \times 10^{-2}$ | link | link ranking | MRR |
| Peptides-func | $15,535$ | $150.9$ | $307.3$ | $1.36 \times 10^{-2}$ | graph | 10-task classif. | AP |
| Peptides-struct | $15,535$ | $150.9$ | $307.3$ | $1.36 \times 10^{-2}$ | graph | 11-task regression | MAE |
| COCO-SP | $123,286$ | $476.9$ | $2,693.7$ | $1.20 \times 10^{-2}$ | node | 81-class classif. | F1 |
| PascalVOC-SP | $11,355$ | $479.4$ | $2,710.5$ | $1.20 \times 10^{-2}$ | node | 21-class classif. | F1 |
| **Open Graph Benchmark** | | | | | | | |
| PCQM4Mv2 | $3,746,620$ | $14.1$ | $14.6$ | $7.25 \times 10^{-2}$ | graph | regression | MAE |
| Molhiv | $41,127$ | $25.5$ | $27.5$ | $4.29 \times 10^{-2}$ | graph | binary classif. | AUROC |
| Molpcba | $437,929$ | $26.0$ | $28.1$ | $4.13 \times 10^{-2}$ | graph | 128-task classif. | AP |
| Code2 | $452,741$ | $125.2$ | $124.2$ | $1.59 \times 10^{-2}$ | graph | 5 token sequence | F1 |
| PPA | $158,100$ | $243.4$ | $2,266.1$ | $3.25 \times 10^{-2}$ | graph | 37-task classif. | Accuracy |

The first group consists of datasets used by GraphGPS (Rampasek et al., 2022) and multiple other work in the community (He et al., 2023; Shirzad et al., 2023). This collection consists of datasets from 5 distinct sources, and we further divide them into 2 groups: Long Range Graph Benchmark (LRGB) (Dwivedi et al., 2022b) and Open Graph Benchmark (OGB) (Hu et al., 2020; 2021). For LRGB datasets, we respect the similar budget of 500k parameters adopted by the previous literature (Dwivedi et al., 2022b).

**Splits.** For these datasets, we employ the experimental setup used in GraphGPS for preprocessing and data splits, please refer to the original work for details (Rampasek et al., 2022).

**Molhiv and Molpcba** are molecular property predictions sourced from the OGB Graph (OGBG) collection (Hu et al., 2020). These predictions are derived from MoleculeNet, and each graph in the dataset represents a molecule with atoms as nodes and chemical bonds as edges. The primary task involves making binary predictions to determine whether a molecule inhibits HIV virus replication or not.

**Code2** is a dataset collection consisting of Abstract Syntax Trees (ASTs) extracted from Python definitions sourced from more than 13,000 repositories on GitHub. This dataset is also part of the OGBG. The primary objective of this task is to predict the sub-tokens that compose the method name. This prediction is based on the Python method body represented by the AST and its associated node features.

**PPA** is a protein association network from OGBG. It is compiled from protein-protein association networks originating from a diverse set of 1,581 species, spanning 37 distinct taxonomic groups. The primary objective of this task is to predict the taxonomic group from which a given protein association neighborhood graph originates.

**PascalVOC-SP and COCO-SP** are node classification datasets included in the LRGB collection. These datasets are derived from super-pixel extraction on PascalVOC (Everingham et al., 2010) and MS COCO datasets (Lin et al., 2014) using the SLIC algorithm. Each super-pixel (node) in these datasets is assigned to a specific class.

***PCQM4Mv2*** is a molecular (quantum chemistry) dataset obtained from the OGB Large-Scale Challenge (LSC) collection, focusing on predicting the DFT (density functional theory)-calculated HOMO-LUMO energy gap of molecules using their 2D molecular graphs—a critical quantum chemical property (Hu et al., 2021). It is important to note that the true labels for the test-dev and test-challenge dataset splits have been kept private by the challenge organizers to avoid any bias in the evaluation process. For our evaluation, we adopted the original validation set as our test set and reserved a random sample of 150,000 molecules for the validation set, following the experimental setting employed in GraphGPS.

**Peptides-func, Peptides-struct, and PCQM-Contact** are molecular datasets from LRGB collection. Peptides-func and Peptides-struct are both derived from $15,535$ peptides retrieved from SAT-Pdb (Singh et al., 2016), but they differ in their task. While Peptides-func is a graph classification task based on the peptide function, Peptides-struct is a graph regression task based on the 3D structure of the peptides. These graphs have relatively large diameters and are constructed in such a way that they necessitate long-range interaction (LRI) reasoning to achieve robust performance in their respective tasks. PCQM-Contact is derived from the PCQM4M (Hu et al., 2021), which includes available 3D structures. It has been filtered to retain only the molecules that were in contact at least once. The main objective of this dataset is to identify, at the edge level, whether two molecules have been in contact.

## A.2  DATASETS WITH SINGLE GRAPH

Table 8: Overview of the graph learning dataset.

| Dataset | # Nodes ($N$) | # Edges ($M$) | Sparsity ($\frac{M}{N^2}$) | # Features | # Classes |
|---|---|---|---|---|---|
| Flickr | $89,250$ | $899,756$ | $1.12 \times 10^{-4}$ | 500 | 7 |
| ogbn-arxiv | $169,343$ | $1,166,243$ | $3.97 \times 10^{-5}$ | 128 | 40 |
| Reddit | $232,965$ | $114,615,892$ | $1.95 \times 10^{-6}$ | 602 | 41 |
| Yelp | $716,847$ | $13,954,819$ | $2.26 \times 10^{-6}$ | 300 | 100 |

The second group comprises large node classification datasets, and we utilize standard accuracy metrics for evaluation, except for Yelp where we use micro-F1 following the general practice. **Splits.** For the following datasets, we use publicly available standard splits across all datasets.

**Reddit** [1] (Hamilton et al., 2017) is a dataset derived from Reddit posts. Each node in the dataset represents a post, and two posts are connected if they have been commented on by the same user. The task is to classify which subreddit (community) a post belongs to.

**Flickr** and ***Yelp*** datasets are obtained from their respective networks (Zeng et al., 2020). In the Flickr dataset, nodes represent uploaded images, and two nodes are connected if they share common properties or attributes. In the Yelp dataset, two nodes are connected if they are considered friends within the social network.

**OGBN- Arxiv** is OGB Node Prediction (OGBN) dataset (Hu et al., 2020) which is a citation network that connects Computer Science papers from Arxiv. The features represent bag-of-word representations of the paper's title and abstract. The task is to identify the area of the papers.

## B  ARCHITECTURE DETAILS

### B.1  RELATIVE ENCODINGS

In Section 3.1, we discuss how to incorporate positional/structural encodings. Importantly, our algorithm does not implicitly retain a dense attention matrix, making the integration of relative encodings more challenging. We work with a relative encoding matrix $U_r \in \mathbb{R}^{N \times N}$, such as adjacency matrix or spatial information matrix, and first create a low-rank approximation (Jolliffe & Cadima, 2016)

---

[1]Reddit dataset is derived from the Pushshift.io Reddit dataset, which is a previously existing dataset extracted and obtained by a third party that contains preprocessed comments posted on the social network Reddit and hosted by pushshift.io.

denoted by $U_r^* \in \mathrm{R}^{N \times d_r}$, where $d_r$ is the rank of the approximation. Subsequently, we append the approximate relative encoding matrix to the node features, and create an updated feature matrix $X^* = [X, U_r^*]$. Note that, both node and edge positional/structural encodings can be extracted offline as a preprocessing step.

## B.2 END-TO-END TRAINING

---
**Algorithm 2** Permute

---
**Input:** Adjacency matrix $\mathbf{A} \in \mathbb{R}^{N \times N}$; Node embeddings $\mathbf{X} \in \mathbb{R}^{N \times d}$; Node labels $\mathbf{Y} \in \mathbb{R}^{N \times d_y}$; Permutation $\mathbf{P} \in \mathbb{R}^{N \times N}$;
  1. $\mathbf{A}' \leftarrow \mathbf{P} \cdot \mathbf{A} \cdot \mathbf{P}^\top$
  2. $\mathbf{X}' \leftarrow \mathbf{P} \cdot \mathbf{X}$
  2. $\mathbf{Y}' \leftarrow \mathbf{P} \cdot \mathbf{Y}$
  Return $\mathbf{A}', \mathbf{X}', \mathbf{Y}'$

---

Algorithm 2 performs a permutation operation on the graph and node features based on the given permutation matrix and returns the permuted node features, labels and adjaceny matrix.

---
**Algorithm 3** Propagate

---
**Input:** Adjacency matrix $\mathbf{A} \in \mathbb{R}^{N \times N}$; Node embeddings $\mathbf{X} \in \mathbb{R}^{N \times d}$;
  1. $\hat{\mathbf{A}} = $ Normalized $\mathbf{A}$
  2. $\mathbf{X}' = \hat{\mathbf{A}}\mathbf{X}$
  3. Return $[\mathbf{X}, \mathbf{X}']$

---

The algorithm 3 outlines the Propagation Block (PB), which plays a key role in the process. The PB starts by aggregating neighborhood embeddings, optionally a normalization can be applied. Normalized $\mathbf{A}$ can be derived in different ways. The standard GCN derives it as follows: $\hat{\mathbf{A}} = \mathbf{D}^{-1/2}\mathbf{A}\mathbf{D}^{-1/2}$, where $\mathbf{D}^{-1/2}$ where $\mathbf{D} = \mathrm{diag}(\mathbf{A1})$, and $\mathbf{1}$ is a column vector of ones. Preserving the original node features after propagation is essential. While some models, like GCN, achieve this by introducing self-edges to the original graph, this approach has a limitation: nodes treat their own embeddings and their neighbors' embeddings equally in terms of importance. To overcome this limitation, we adopt a different strategy. We concatenate the original node embeddings with the propagated embeddings, similar to a dense residual connection (Huang et al., 2017). Notably, this step does not involve any learnable parameters. In Section 4.3, we explore several other variants with GNN models.

---
**Algorithm 4** Projection

---
**Input:** Node embeddings $\mathbf{X} \in \mathbb{R}^{N \times d}$;
  1. In parallel across $N$: $Z = \mathsf{Linear}(X)$, $\mathsf{Linear} : \mathbb{R}^d \to \mathbb{R}^{(K+1)d}$
  3. Reshape and split $Z$ into $X_1, X_2, \ldots, X_K, V$, where $X_k, V \in \mathbb{R}^{d \times N}$
  Return $X_1, X_2, \ldots, X_K, V$

---

---
**Algorithm 5** Forward pass of GECO

---
**Input:** Adjacency matrix $\mathbf{A} \in \mathbb{R}^{N \times N}$; Node embeddings $\mathbf{X} \in \mathbb{R}^{N \times d}$;
  1. $\mathbf{X} = BatchNorm(Propagate(\mathbf{X}, \mathbf{A}))$
  2. $\mathbf{X} = GCB(\mathbf{X}, \mathbf{A})$
  Return $\mathbf{X}$

---

## C COMPUTATIONAL COMPLEXITY DISCUSSION

Table 9 presents an overview of the complexities associated with various models. It is important to note that while we provide specific categorizations, certain models within those categories may exhibit different complexities. Hence, the complexities presented here represent the general case.

| Model | GNN | Dense GT | Layer GT | GraphGPS | GECO |
|---|---|---|---|---|---|
| Long-range Modeling | × | ✓ | × | ✓ | ✓ |
| Time | $\mathcal{O}(L(N+M))$ | $\mathcal{O}(LN^2)$ | $\mathcal{O}(L(N+M)+L^2)$ | $\mathcal{O}(LN^2)$ | $\mathcal{O}(L(N\log N+M))$ |
| Memory | $\mathcal{O}(L(N+M))$ | $\mathcal{O}(LN^2)$ | $\mathcal{O}(L(N+M))$ | $\mathcal{O}(LN^2)$ | $\mathcal{O}(L(N\log N+M))$ |

Table 9: Computational Complexity Comparison for Full Batch Training

## C.1 Related Work

**GNN** models (Kipf & Welling, 2017; Hamilton et al., 2017) can be evaluated efficiently using sparse matrix operations in linear time with the number of nodes and edges in the graph. Although, some other models such as Graph Attention Networks (GAT) (Veličković et al., 2018) and its variant GATv2 (Xu et al., 2019) which we categorize under *Sparse GT* at Section 2.2 has higher complexity, due to the attention mechanism. Asymptotically, these models can achieve higher computational efficiency compared to other methods such as Dense GT and Layer GT. Furthermore, they encompass a rapidly expanding line of research that investigates mini-batch sampling methods on GNNs (Hamilton et al., 2017; Zou et al., 2019; Zeng et al., 2020; 2021; Balın & Çatalyürek, 2023).

**Dense GT** (Dwivedi & Bresson, 2020; Kreuzer et al., 2021; Ying et al., 2021; Mialon et al., 2021; Chen et al., 2022) involves pairwise attention between every node regardless of the connectivity of the graph, and hence has quadratic complexity with the number of nodes $\mathcal{O}(N^2)$. Recently, a variant of DenseGT called Relational Attention has been introduced, which involves additional edge updates, further increasing the overall complexity to $\mathcal{O}(N^3)$. Given that standard mini-batching methods cannot be applied to DenseGT, its application is limited to small datasets.

**Layer GT.** We categorize NAGphormer (Chen et al., 2023) under this category. NAGphormer employs attention on the layer (hop) tokens and exhibits a complexity similar to GNNs, given that the number of layers is typically fixed and smaller compared to the number of nodes or edges. As discussed in detail in Section E.4, while NAGphormer may not completely address issues inherited in GNNs, it offers several highly desirable properties. Firstly, it performs feature propagation as a preprocessing step, enabling the results to be reused. Additionally, during training, NAGphormer does not require consideration of the connectivity, which allows it to leverage off-the-shelf traditional mini-batch training methods, thereby achieving parallelism and scalability on large datasets.

**Hybrid GT** includes recently proposed GraphGPS framework (Rampasek et al., 2022). GraphGPS combines the output of a GNN Module and Attention Module, and outputs of the two module is summed up within each layer. We discuss the differences between GECO and GraphGPS in detailed in Appendix E.3. The complexity of GraphGPS primarily depends on its attention module, which acts as a bottleneck. Despite offering subquadratic alternatives for attention, the reported results indicate that the best performance is consistently achieved when using the Transformer as the attention module. Consequently, similar to Dense GT, GraphGPS exhibits a complexity of $\mathcal{O}(N^2)$, making it less suitable for large datasets. In essence, GraphGPS serves as a versatile framework for combining GNN and attention, along with their respective alternatives.

**Positional Encodings (PE)** play a crucial role in various tasks, and while some, like node degrees used in Graphormer (Ying et al., 2021), can be efficiently computed, others, such as Laplacian eigen vectors, Laplacian PE, or all pairs shortest path (for relative PE), involve computationally expensive operations, usually $\mathcal{O}(N^3)$ or higher. The good news is step can be computed once as a preprocessed step and does not necessarily be computed in GPU. Nevertheless, for extremely large graphs, computing PE can still be computationally infeasible.

## C.2 GECO

**Proposition 1** *Local Propagation Block can be computed* $\mathcal{O}(N+M)$ *using Sparse Matrix Matrix (SpMM) multiplication between* $X^{(l)}$ *and* $A$ *in linear time complexity, where* $M = |E|$.

**GECO** is composed of two building blocks: Local Propagation Block (LCB) and the Global Context Block (GCB). In Proposition 1, we discuss the complexity of LCB, which is $\mathcal{O}(N+M)$. This step exhibits a similar memory complexity of $\mathcal{O}(N+M)$.

**Proposition 2** *GCB can be computed $\mathcal{O}(N \log N)$ by using Fast Fourier Transform (FFT).*

### C.3 END-TO-END TRAINING

Next, we analyze the complexity of GCB in Proposition 2, which is $\mathcal{O}(N \log N)$. In the paper, we specifically focus on the case when the GCB recurrence order is set to 2. However, we can generalize this to $K$ recurrence, from which we derive the following complexity components:

1. Each GCB includes $(K+1)$ linear projections, resulting in a complexity of $\mathcal{O}((K+1)N)$.

2. Next, we have $K$ element-wise gating operations, contributing a complexity of $\mathcal{O}(KN)$.

3. Finally, there are $K$ FFT convolutions, where both the input and filter sizes are $N$, resulting in a complexity of $\mathcal{O}(KN \log N)$.

As a result, the generalized complexity of GCB can be expressed as $\mathcal{O}(KN \log N)$. Considering the end-to-end training complexity of GECO, we can combine the complexities of LCB and GCB, resulting in $\mathcal{O}(KN \log N + M)$.

Next, we examine the memory complexity of GECO, with a particular focus on the FFT convolutions used. The standard PyTorch (Paszke et al., 2019) FFT Convolution typically requires $\mathcal{O}(N \log N)$ memory. However, it is possible to optimize this complexity to $\mathcal{O}(N)$ by leveraging fused kernel implementations of FFT Convolutions (Fu et al., 2023a). As a result, we can express GECO's memory complexity as $\mathcal{O}(KN + M)$ when utilizing these fused FFT Convolution implementations, where $K$ is the recurrence order.

## D EXPERIMENTAL DETAILS

### D.1 BASELINES

Our baselines through Tables 2 to 5 include GCN (Kipf & Welling, 2017), Graphormer (Ying et al., 2021), GIN (Xu et al., 2019), GAT (Veličković et al., 2018), GatedGCN (Bresson & Laurent, 2017; Dwivedi et al., 2023), PNA (Corso et al., 2020), DGN (Beaini et al., 2021), CRaW1 (Toenshoff et al., 2021), GIN-AK+ (Zhao et al., 2022), SAN (Kreuzer et al., 2021), SAT (Chen et al., 2022), EGT (Hussain et al., 2022), GraphGPS (Rampasek et al., 2022), and Exphormer (Shirzad et al., 2023).

Our baselines for Table 4 include GCN (Kipf & Welling, 2017), GraphSAGE (Hamilton et al., 2017), GraphSaint (Zeng et al., 2020), ClusterGCN (Chiang et al., 2019), GAT (Veličković et al., 2018), JK-Net (Xu et al., 2018), GraphGPS (Rampasek et al., 2022) and Exphormer (Shirzad et al., 2023), Graphormer (Ying et al., 2021), SAN (Kreuzer et al., 2021), SAT (Chen et al., 2022), ANS-GT (Chien et al., 2021), GraphGPS (Rampasek et al., 2022), HSGT (Zhu et al., 2023).

### D.2 IMPLEMENTATION AND COMPUTE RESOURCES

**Implementation.** We have implemented our model using PyTorch-Geometric (Fey & Lenssen, 2019), GraphGPS (Rampasek et al., 2022) and Safari libraries (Poli et al., 2023). For the evaluation at Section 4.1, we have integrated our code into the GraphGPS framework as a global attention module. For the evaluation at Section 4.2, we have implemented our own framework to efficiently run large datasets [2].

**Compute Resources.** We have conducted our experiments on an NVIDIA DGX Station A100 system with 4 80GB GPUs.

### D.3 HYPERPARAMETERS

**Baselines.** For the baseline results presented at Tables 2 to 5 we reuse the results reported by GraphGPS and Exphormer (Rampasek et al., 2022; Shirzad et al., 2023). Moreover, for the baseline

---

[2] Our implementations will be open-sourced during or after the double-blind review process

results presented at Table 4, we present the previously reported results in the literature (Han et al., 2023; Shirzad et al., 2023; Zeng et al., 2021; Zhu et al., 2023).

**GECO.** For datasets at Table 7, we drop in and replace the global attention module with GECO. The missing results are marked by $-$. Our choice of hyperparameters is guided by GraphGPS and Exphormer (Rampasek et al., 2022; Shirzad et al., 2023). We started with the hyperparameters recommended by the related work including optimizer configurations, positional encodings, and structural encodings. Then we proceed to hand-tune some optimizer configurations, dropout rates, and hidden dimensions by simple line search by taking validation results into account. On multiple datasets including PascalVOC, COCO, molpcha, and code2, we have eliminated the positional and structural encodings.

For datasets at Table 8, we have used hyperparameter optimization framework Optuna (Akiba et al., 2019) with Tree-structured Parzen Estimator algorithm for hyperparameter suggestion with each tuning trial using a random seed. We reported the test accuracy achieved with the best validation configuration over 10 random seeds. As part of our public code release, we will provide all configuration files detailing our hyperparameter choices.

# E ADDITIONAL EXPERIMENTS DISCUSSIONS

## E.1 EXTENDED RUNTIME STUDY

In this subsection, we provide details on runtime ablation study in Section 4.3.

**Experimental Setting.** We have generated random graphs using Erdős-Rényi model. We increased the number of nodes from 512 to 4.2 million by doubling the number of nodes at consecutive points, using a total of 14 synthetic datasets. Furthermore, we set the sparsity factor of each graph to $10/N$, where N is the number of nodes as defined before, aligning the sparsity of the graph with that of large node prediction datasets in Table 8. Additionally, we fixed the number of features at 108 across all datasets. We utilized publicly available FlashAttention implementation (Dao et al., 2022)[3]. For FlashAttention, we used 4 heads. In both GECO and FlashAttention, the number of hidden units is set as the number of features.

Table 10: Runtime Comparison of GECO and FlashAttention (Dao et al., 2022) on synthetic datasets. $\mathcal{O}(N^2/(N \log N)) = \mathcal{O}(N/ \log N)$ characterizes the speedup. Sparsity factor of each graph is set as $10/N$.

| N | GECO (ms) | FlashAttention (ms) | Relative Speedup |
|---|---|---|---|
| 512 | 1.88 | 0.27 | 0.14 |
| 1,024 | 2.13 | 0.32 | 0.15 |
| 2,048 | 2.11 | 0.31 | 0.15 |
| 4,096 | 2.42 | 0.32 | 0.13 |
| 8,192 | 2.12 | 0.51 | 0.24 |
| 16,384 | 2.13 | 1.84 | 0.86 |
| 32,768 | 2.63 | 6.92 | 2.63 |
| 65,536 | 3.73 | 28.74 | 7.70 |
| 131,072 | 6.21 | 115.23 | 18.56 |
| 262,144 | 15.74 | 458.64 | 29.14 |
| 524,288 | 41.72 | 1830.29 | 43.87 |
| 1,048,576 | 83.90 | 7317.04 | 87.21 |
| 2,097,152 | 173.15 | 29305.77 | 169.25 |

The results demonstrate that as the number of nodes grows larger, GECO achieves significant speedups with respect to FlashAttention. This is anticipated due to GECO's complexity of $\mathcal{O}(N \log N + M)$, while attention's complexity is $\mathcal{O}(N^2)$. Considering the sparsity of real-world datasets, $N$ becomes a dominant factor, leading to a speedup characterized by $\mathcal{O}(N/ \log N)$.

In summary, our findings support GECO's efficiency for larger-scale applications, whereas for smaller scales, the choice between the two could be influenced by factors beyond just performance.

---

[3]https://github.com/Dao-AILab/flash-attention

Table 11: Comparison with various Graph Methods reported by (Min et al., 2022).

| Model | | molhiv | molpcba | Flickr | ogbn-arxiv |
|---|---|---|---|---|---|
| | | ROC-AUC↑ | AP↑ | Acc↑ | Acc↑ |
| TF | vanilla | 0.7466 | 0.1676 | 0.5279 | 0.5598 |
| GA | before | 0.7339 | 0.2269 | 0.5369 | 0.5614 |
| | alter | 0.7433 | 0.2474 | **0.5374** | 0.5599 |
| | parallel | **0.7750** | 0.2444 | **0.5379** | 0.5647 |
| PE | degree | 0.7506 | 0.1672 | 0.5291 | 0.5618 |
| | eig | 0.7407 | 0.2194 | 0.5278 | **0.5658** |
| | svd | 0.7350 | 0.1767 | 0.5317 | **0.5706** |
| AT | SPB | 0.7589 | **0.2621** | 0.5368 | 0.5605 |
| | PMA | 0.7314 | 0.2518 | 0.5288 | 0.5571 |
| | Mask-1 | **0.7960** | **0.2662** | 0.5300 | 0.5598 |
| | Mask-n | 0.7423 | 0.2619 | 0.5359 | 0.5603 |
| GECO (Ours) | | **0.7980 ± 0.0200** | **0.2961 ± 0.0008** | **0.5555 ± 0.0025** | **0.7310 ± 0.0024** |

As discussed throughout Section 1-Section 4, the scalability gains are not expected for small graphs. This is mostly due to low hardware utilization incurred by available FFT implementations. However, GECO still remains a promising approach due to its high prediction quality, as we demonstrated in Section 4.1. On the other hand, on larger graphs, GECO exhibits significant scalability, as demonstrated in Section 4.2. It consistently outperforms dense GTs on all large datasets and remains superior or competitive when compared to the orthogonal approaches.

## E.2 COMPARISON WITH VARIOUS GRAPH TRANSFORMERS VARIANTS

The survey (Min et al., 2022) classifies existing methods for enhancing Transformer's awareness of topological structures into three main categories: 1) Integrating GNNs as auxiliary modules (GA), (2) Enhancing positional embeddings from graphs (PE), and (3) Improving attention matrices using graphs (AT).

Regarding GA, the survey explores different approaches to combining off-the-shelf GNNs with Transformer. These methods involve adapting a series of GNNs and then applying a series of Transformers sequentially (before) (Wu et al., 2021), integrating GNNs and Transformers consecutively (Alternatively) (Lin et al., 2021), or utilizing them in parallel at each layer (Rampasek et al., 2022). Notably, these models straightforwardly merge pre-existing GNN and Transformer models, resulting in separate parameters and intermediate non-linearities for each module, with independently applied skip connections.

However, GECO does not precisely align with the taxonomy defined in the survey (Min et al., 2022). In GECO, we did not just use LCB as an auxiliary module to Transformer. Instead, we designed a new compact layer comprising local and global blocks. We eliminated the intermediate non-linearities and parameters to reduce the overall number of parameters, simplifying the training process. We applied skip connections to the entire GECO layer as a whole, rather than separately. These deliberate design choices distinguish GECO from the use of off-the-shelf methods. Please also refer to Appendix E.3 for further design details.

Furthermore, Table 11 highlight that GECO achieves consistent superior predictive quality across all datasets. Specifically, on arxiv and molphcba, GECO achieves significant relative improvements of up to 28.11% and 11.23%, respectively. We also note that, PE and AT are orthogonal approaches to our work.

## E.3 COMPARISON BETWEEN GECO AND GRAPHGPS

As GECO can be considered a hybrid method, a natural question arises regarding what sets it apart from GraphGPS. In this section, we delve into the distinctions between these two models. The fundamental difference between GECO and GraphGPS layers lies in their design as illustrated in

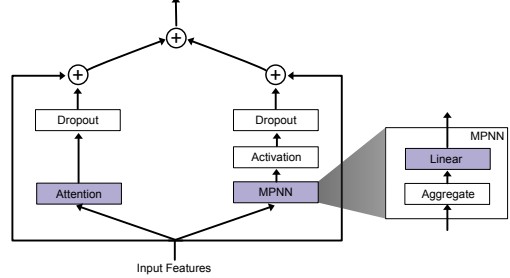 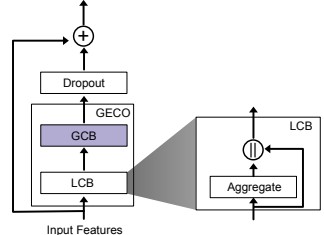

(a) **GraphGPS Layer** consists of an MPNN and global attention modules with each module having its own skip connections and optional dropout. The modules are evaluated in parallel and summed up at the end. Both MPNN and attention modules usually have learnable weights.

(b) The **GECO Layer** comprises Local Propagation Block (LCB) and Global Context Block (GCB), evaluated sequentially with a skip connection across the entire block. Notably, LCB lacks learnable weights, serving as a pre-step to GCB, which incorporates the learnable weights.

Figure 3: A comparison between GraphGPS and GECO, where the layers with learnable weights are highlighted in color.

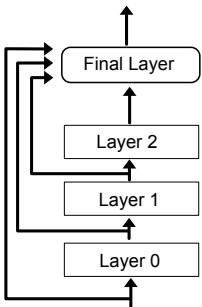

Figure 4: Illustration of JK-Nets with 3 layers. It is important to note that the Final Layer can be implemented using different layers, and it does not necessarily have to be the same as the intermediate layers. Although the original work While the original work (Xu et al., 2018) did not introduce a dense skip connection from the original inputs to the Final Layer, we have included it here for the sake of consistency in notation.

Figure 3. Given input features, the MPNN and attention modules of GraphGPS are evaluated in parallel and then summed up. Each module encompasses its own set of learnable weights, activation functions, dropout layers, and residual connections. On the other hand, in the case of GECO, LCB and GCB are evaluated sequentially. Notably, LCB does not incorporate any learnable weights, activation functions, or dropout layers; it functions as a feature propagation block with a dense skip connection. Instead, GECO's weights are encapsulated within the GCB block. Furthermore, the residual connection is applied to the entire GECO block. In practice, GECO and GraphGPS can be combined in various ways. In our experiments, we chose to employ GECO as an attention module to facilitate a direct comparison with GraphGPS and Exphormer. However, it is possible to integrate GraphGPS and GECO differently. Potential options include substituting GCB with a Transformer or replacing LCB with an MPNN.

### E.4 CONNECTION BETWEEN JUMPING KNOWLEDGE NETWORKS AND NAGPHORMER

Jumping Knowledge Networks (JK-Nets) (Xu et al., 2018) have been introduced as GNNs with a variant of dense skip connection. The difference from the original dense skip connections (Huang et al., 2017) is that instead of establishing dense connections between every consecutive layer, JK-Nets establish dense skip connections from each layer to a final aggregation layer as illustrated in Figure 4. Given this framework, we can recover NAGphormer as a special case of JK-Nets with two simple configurations:

1. Replace non-linear transformation function of the GNN with the identity function. That is we do not use learnable weights, simply utilize traditional feature propagation.

2. Set Final-Layer as multi-head attention.

With these two simple modifications, one can recover the NAGphormer as a specific instance of JK-Nets. Here NAGphormer corresponds to an MHA layer with tokens produced by a GNN with no learnable weights, or traditional feature propagation.

