# OpenReview forum: "A Fast and Effective Alternative to Graph Transformers"
_ICLR.cc/2024/Conference — Submitted to ICLR 2024_

### Official Review · Reviewer_Utmf · 2023-10-26

**Soundness:** 2 fair
**Presentation:** 2 fair
**Contribution:** 2 fair
**Rating:** 5
**Confidence:** 5

**Summary:**

This paper focuses on a good question, i.e., the scalability of Graph Transformers (GTs). GTs suffer from quadratic complexity when the node number in a certain graph is very large. The authors propose a somewhat improvement of GTs. An improved (the authors define it as an alternative) dense attention mechanism is utilized to reduce the computing complexity of GTs. It is claimed that the proposed GECO can capture long-range dependencies. The proposed method shows some improvements in a limit number of datasets. The appendix is of a lot content including details of experimental settings, related work, a brief discussion of computational complexity discussion, etc.

**Strengths:**

-This paper focuses on a good and significant research question and poses a very smart improvement by modifying dense attention mechanisms of GTs.

-This paper includes rich contents. An additional appendix containing many details that can help to clearly understand the paper.

-The proposed method shows some acceptable experimental results when comparing with baselines and good ablation experiments.

**Weaknesses:**

-The innovation is not strong enough, which diminishes the significance of the paper. Essentially, the authors replaced the multi-head self-attention module in the original Transformer with a global convolutional module, and then they claim their proposal is to improve efficiency. However, the necessity of this replacement needs to be considered, and it appears to be of limited significance. The primary issue lies in the absence of self-attention mechanism, resulting in a diminished capability to capture long-range dependencies. Experimental results on large datasets, such as PCQM and COCO, indicate that the model's performance is inferior to other Graph Transformer methods.

-Lacking experimental results to verify “fast” of the proposed method. Specifically, there is no emphasis in the experimental results, no parameter complexity analysis, no comparison of computing resource consumption or computing time. These are fundamental experiments in verifying “fast” of a certain method. And the results provided to demonstrate the 'effectiveness' of the proposed method in capturing long-distance dependencies, as shown in Tables 1, 2, and 4, may not offer sufficiently strong evidence for its superior performance. Overall, the title of this paper is ambitious and likely to capture attention with insufficient innovative approach, even though the authors claimed “they are the first to”.

-The design, organization, and writing of this paper are not very clear to me. Firstly, the motivation seems to enhance GT, but the authors care a lot about capturing long-range dependencies, which I have illustrated in last point, the results are not impressive enough. If the authors want to show the outperformance of trade-off between capturing long-range dependencies and fast calculation/computation, there is a lack of comparison of baselines including those methods not using Transformers. Then, If the authors want to show the improvement of the enhanced GT in effectiveness, the results are not competitive. And I think the authors also need to refer to some recent studies such as “Hierarchical Transformer for Scalable Graph Learning”. Next, the authors aim to illustrate that their proposed method is fast and has distinct difference from GraphGPS. But why GraphGPS? It confuses me.

**Questions:**

-What exact problem the authors want to solve? And how you directly verified that the problem is well solved, with what metric/way?

-How to balance the trade-off between fast and efficiency? Why is your method the best?

-I am well aware that the comparison of computation complexity (theoretically) among several models including GECO. But what about the experimental verification?

---

> ### Author Response · Authors · 2023-11-17
>
> Many thanks for a thorough and insightful review. Below, we address each concern, organized to avoid repetitive discussions
>
> > @W3 "Next, the authors aim to illustrate that their proposed method is fast and has distinct difference from GraphGPS. But why GraphGPS? It confuses me."
>
> We understand your confusion. Let us clarify it by reviewing the literature chronologically.
>
> 1. Graph Transformer has been proposed, suggesting that attention should only be applied to connected nodes, consequently utilizing the adjacency matrix as an attention mask [1].
> 2. GraphTrans [2] and Graphormer [3] have been concurrently proposed, utilizing a self-attention mechanism, demonstrating that attention can be applied to the entire set of nodes and significantly improve predictive performance. Specifically, Graphormer achieved impressive results by incorporating several effective structural encodings.
> 3. Many GTs have been proposed, many of which are approaches proposing different structural/positional encodings, extracting additional substructure tokens, or applying attention to specific substructures centered around a node, which we detail in our manuscript. Many of these approaches are orthogonal approaches and they utilize self-multi-head attention as their kernel.
> 4. GraphGPS [4] has been proposed, combining GNN and attention modules side by side at each layer, offering recipes for off-the-shelf GNN, attention and positional/structural encodings combinations, and achieving state-of-the-art results across many graph datasets.
> 5. **We proposed GECO as a compact layer consisting of local propagation and global convolution model blocks.** As both GECO and GraphGPS utilizes local and global blocks, we clarify their distinction to address potential reader questions. Unlike previous methods, in GECO, we designed a compact layer comprising local and global modules without intermediate parameters and non-linearities, applying skip connections to the layer as a whole. Please refer to our response to R-XpCe @W1 for more details. In addition, our work can complement other GT approaches focusing on various positional/structural encodings, or clustering/coarsening/substructure-based strategies.
>
> > @W3 "If the authors want to show the outperformance of trade-off between capturing long-range dependencies and fast calculation/computation, there is a lack of comparison of baselines including those methods not using Transformers"
> @W3 "And I think the authors also need to refer to some recent studies such as “Hierarchical Transformer for Scalable Graph Learning”
>
> We have added the Hierarchical Transformer for Scalable Graph Learning (HSGT) [5] along with additional baselines from [5] to our revised manuscript. Below, we summarize the comparison between the two works on common datasets:
>
>
> | Model         | Flickr  | Yelp   | Arxiv   |
> |---------------|:------------:|:--------:|:----------------:|
> | Graphormer [5] | OOM | OOM | OOM |
> | Graphormer-SAMPLE [5] | 51.93 ± 0.21 | 60.01 ± 0.45  | 70.43 ± 0.20  |
> | SAN [5] |  OOM | OOM | OOM |
> | SAT [5] |  OOM | OOM | OOM |
> | GraphGPS (Transformer) |  OOM | OOM | OOM |
> | SAT-SAMPLE [5] | 50.48 ± 0.34 | 60.32 ± 0.65 | 68.20 ± 0.46  |
> | Exphormer [6]| NA | NA | 72.44 ± 0.28 |
> | ANS-GT [5] | NA | NA | 68.20 ± 0.46|
> | HSGT [5] | 54.12 ± 0.51        | **63.47 ± 0.45**          | 72.58 ± 0.31             |
> | GECO   | **55.55 ± 0.25**         | 63.18 ± 0.59        | **73.10 ± 0.24**           |
>
> - **HSGT and GECO are orthogonal approaches.** HSGT's attention components could be swapped with GECO's for a hierarchical graph learning model using global convolutions, offering a potential future direction. GECO provides kernel-wise scalability in place of self-multi-head attention, while HSGT achieves structural scalability through graph partitioning and coarsening.
> - **The comparison shows competitive performance:** GECO outperforms HSGT on Flickr and Arxiv, while HSGT surpasses GECO on Yelp.
> - **GECO vs Dense GTs (Graphormer/Graphormer-Sample/GraphGPS)**: While Graphormer and GraphGPS, which rely on self-attention, encounters out-of-memory issues in every scenario, GECO effectively scales to these datasets. Furthermore, Graphormer with sampling falls short in achieving competitive quality when compared to GECO across all datasets.
>
> Additionally, Reddit dataset comes in multiple variants. In our study, we used the variant originally proposed in [7] available in PyG [8]. However, we noticed that HSGT uses a different variant. Below, we summarize the difference.
>
>
> | Dataset       |      N   (Ours)    |    M  (Ours)   |    N  (HSGT [5])    |    M (HSGT [5])  |
> |---------------|:------------:|:--------:|:----------------:|:----------------:|
> | Reddit        |  233K     |114.6 M|  233K |  11.6M |

---

> ### Author Response · Authors · 2023-11-17
>
> >@W1 " However, the necessity of this replacement needs to be considered, and it appears to be of limited significance. The primary issue lies in the absence of self-attention mechanism, resulting in a diminished capability to capture long-range dependencies."
> @W1 "Experimental results on large datasets, such as PCQM and COCO, indicate that the model's performance is inferior to other Graph Transformer methods."
> @W2 "And the results provided to demonstrate the 'effectiveness' of the proposed method in capturing long-distance dependencies, as shown in Tables 1, 2, and 4, may not offer sufficiently strong evidence for its superior performance."
> @W3 "Firstly, the motivation seems to enhance GT, but the authors care a lot about capturing long-range dependencies, which I have illustrated in last point, the results are not impressive enough."
> @W3 "If the authors want to show the improvement of the enhanced GT in effectiveness, the results are not competitive."
>
> We understand that the reviewer's main concern lies around understanding capability of GECO as a replacement for self-attention.
>
> First and foremost, we emphasize that if the concern is to make a direct comparison between self-attention and GECO without using heuristics, one should compare the GECO and GraphGPS rows for a fair comparison in Tables 1, 2 and 3. Next, in the table below, we summarize (1) relative improvements w.r.t. GraphGPS's self attention and (2) relative improvements w.r.t. best GT baselines including orthogonal approaches (with available results).
>
> We highlight empirical evidence showing that replacing self-attention with GECO does not diminish performance on most datasets; in fact, it often leads to improvements:
>
> - **GECO has superior predictive quality on 8 out of 10 datasets w.r.t. GraphGPS with self-attention (Table 1, 2 and 3).**  Specifically, on the first five columns (long-range benchmark datasets), GECO achieves relative improvements of 12.53%, 6.31%, 1.44%, and 5.77%, with a -2.20% decrease in only on COCO-SP. For datasets in columns 6-10, GECO achieves significant relative improvements of by up to 10.36% on 4 datasets, with only a -0.41% decrease only on ogbg-ppa.
>
>
> - **GECO achieves superior quality to Dense GTs utilizing self attention (Graphormer/Graphormer-sample/GraphGPS) on 12 out of 14 datasets (Table 1, 2, 3 and 4).**  Please see table below and also refer to previous discussion and its table. Empirical evidence further supports that GECO does not have diminishing capability as a replacement GT's self-attention; in fact, it leads to improvements.
>
> - **GECO has superior predictive quality on 9 out of 14 datasets w.r.t. across available GT Baselines including orthotogonal ones(Table 1, 2, 3 and 4).**  Please see below.
>
>
> | Model           | PascalVOC-SP | COCO-SP | Peptides-func| Peptides-struct| PCQM-Contact  |ogbg-molhiv  | ogbg-molpcba  | ogbg-ppa| ogbg-code2 | PCQM4Mv2 | Flickr | Reddit | Yelp | Arxiv |
> |-----------------|-------------------------|--------------------|--------------------|------------------------|---------------------|-----------------------|------------------------------|----------------------|------------------------|------------------------|--------------------|--------------------|------------------------|------------------------|
> || F1 score ↑ | F1 score ↑ | AP ↑ |  MAE ↓ | MRR ↑ |AUROC ↑ |  Avg. Precision ↑ |Accuracy ↑ | F1 score ↑ | MAE ↓  |Accuracy ↑ |Accuracy ↑|Accuracy ↑|Accuracy ↑|
> | GraphGPS             | 0.3748 ± 0.0109         | 0.3412 ± 0.0044    | 0.6535 ± 0.0041    | 0.2500 ± 0.0005        | 0.3337 ± 0.0006     |0.7880 ± 0.0101       | 0.2907 ± 0.0028              | 0.8015 ± 0.0033      | 0.1894 ± 0.0024        | 0.0938     | OOM | OOM | OOM | OOM |
> | Best GT  (Model Name)  | 0.3975 ± 0.0037 (Exphormer)         | 0.3455 ± 0.0009 (Exphormer)    | 0.6535 ± 0.0041 (GraphGPS)    | 0.2481 ± 0.0007 (Exphormer)       | 0.3637 ± 0.0020 (Exphormer)     |  0.7880 ± 0.0101 (GraphGPS)       |  0.2907 ± 0.0028 (GraphGPS)              | 0.8015 ± 0.0033 (GraphGPS)      | 0.1937 ± 0.0028 (K-Subtree SAT)      | 0.0858 (GraphGPS)   | 54.12 ± 0.51 (HSGT) | NA | 63.47 ± 0.45 (HSGT) | 72.58 ± 0.31 (HSGT) |
> | GECO (Ours)    | 0.4210 ± 0.0080         | 0.3320 ± 0.0032    | 0.6975 ± 0.0025    | 0.2464 ± 0.0009        | 0.3526 ± 0.0016     | 0.7980 ± 0.0200       | 0.2961 ± 0.0008              | 0.7982 ± 0.0042      | 0.1915 ± 0.002         | 0.08413     | 55.55 ± 0.25 | 96.65 ± 0.05 | 63.18 ± 0.59 | 73.10 ± 0.24 |
> | Relative Improvement w.r.t. GraphGPS with self-attention (%)|  12.53%    |  -2.20%  | 6.31% | 1.44%   | 5.78%  | 1.27% |  1.85%   | -0.41% |  1.11%  |  10.36%  | NA | NA | NA | NA |
> | Relative Improvement w.r.t. Best GT Baseline (%)|  5.91%    |  -3.91%  | 6.31% | 0.69%   | -3.05%  | 1.27% |  1.85%   | -0.41% |  -1.36%  |  10.36%  | 2.64% | NA | -0.46 | %0.72 |
>
> Please refer to the *Dataset Clarification* on the general discussion regards to definition of large graphs in our context.

---

> ### Author Response · Authors · 2023-11-17
>
> > @W2 "Lacking experimental results to verify “fast” of the proposed method. Specifically, there is no emphasis in the experimental results, no parameter complexity analysis, no comparison of computing resource consumption or computing time. These are fundamental experiments in verifying “fast” of a certain method."
> @W2 "Overall, the title of this paper is ambitious and likely to capture attention with insufficient innovative approach, even though the authors claimed “they are the first to”.
> @Q3 "I am well aware that the comparison of computation complexity (theoretically) among several models including GECO. But what about the experimental verification?"
>
> We have taken this concern to heart, recognizing its significance, particularly given the title of our paper, "fast". Please see the general discussion and revised manuscript.
>
> Additionally, we imposed the same parameter budget constraints as previous research did for the datasets listed in Tables 1, 2, and 3. We included these details in the revised manuscript. Table 3 demonstrates that GECO outperforms the GraphGPS self-attention variant by a significant margin, using 8 times fewer parameters on PCQM4Mv2. When both models use the same number of parameters, GECO achieves a 12.53% relative improvement over GraphGPS. Furthermore, GECO's parameter usage is comparable to that of some traditional GNN architectures.
>
> > @Q1 "What exact problem the authors want to solve? And how you directly verified that the problem is well solved, with what metric/way?"
>
> Our primary motivation is to replace computationally intensive self-attention mechanism within graph transformers with a more efficient operator, while not sacrificing the prediction quality. To this end, we designed a compact layer combining local propagation and global convolutions. Our evaluation includes 3 steps to verify our approach:
>
> 1. **Verify competitive quality on commonly used Graph Transformer Datasets**: Please see discussions and tables above.
>
> 2. **Verify scalability and quality to larger node prediction datasets**: Please see discussions and tables above.
>
> 3. **Verify Runtime Efficiency**: In our revised version, we have included runtime efficiency benchmarks. We appreciate reviewers' feedbacks for encouraging us to include this study.
>
>
> > @Q2 "How to balance the trade-off between fast and efficiency? Why is your method the best?"
>
> There are multiple ways to enhance the scalability and efficiency of Graph Transformers, such as coarsening (HSGT [5]), expander graphs (Exphormer [6]) or sampling strategies [5]. Our work, GECO, takes another direction by directly replacing the self-attention kernel. Unlike prior attempts that compromised predictive quality with self-attention approximations [4], GECO maintains robust performance while improving scalability. Moreover, its design is orthogonal with existing scalability methods like HSGT, offering a promising future research direction. Please also see the discussion above.
>
> > @W1 "The innovation is not strong enough, which diminishes the significance of the paper."
>
> Please refer to general discussion.
>
> [1] A Generalization of Transformer Networks to Graphs, AAAI Workshop'21
>
> [2] Representing Long-Range Context for Graph Neural Networks with Global Attention, NeurIPS'21
>
> [3] Do transformers really perform badly for graph representation?, NeurIPS'21
>
> [4] GraphGPS: General Powerful Scalable Graph Transformers, NeurIPS'22
>
> [5] Hierarchical Transformer for Scalable Graph Learning, IJCAI'23
>
> [6] EXPHORMER: Sparse Transformers for Graphs, ICML'23
>
> [7] Inductive Representation Learning on Large Graphs, NeurIPS'17
>
> [8] Fast Graph Representation Learning with PyTorch Geometric, ICLR Workshop'19, [Reddit Dataset Link](https://pytorch-geometric.readthedocs.io/en/latest/generated/torch_geometric.datasets.Reddit.html#torch_geometric.datasets.Reddit)

---

> > ### Comment · Reviewer_Utmf · 2023-11-22
> >
> > I appreciate the authors' efforts in expanding the experimental results, which have helped to address several of my concerns, such as runtime efficiency, and less # parameters. To reflect this, I have revised my score to 5.

---

> > > ### Author Response · Authors · 2023-11-22
> > >
> > > Dear Reviewer Utmf,
> > >
> > > Thank you for your revision! If you have any additional questions or concerns for further clarifications or improvements, please let us know.

---

### Official Review · Reviewer_vPHt · 2023-10-29

**Soundness:** 3 good
**Presentation:** 2 fair
**Contribution:** 3 good
**Rating:** 6
**Confidence:** 3

**Summary:**

This paper targets the quadratic complexity issue in training graph transformers with full-attention over large graph datasets, and proposes GECO, which is a Hyena-based operator that captures both local and global dependencies to replace the original attention operator. The authors conduct extensive experiments in demonstrating GECO’s effectiveness over long-range and large graph datasets. In addition, the authors empirically demonstrate GECO’s insensitivity to node ordering wrt. performance.

**Strengths:**

1.	The targeted quadratic complexity issue reside in graph transformers is meaningful. The designed GECO not only makes it sub-quadratic, but also remains in a considerable performance level.
2.	The experiments are conducted extensively. The results seem promising.

**Weaknesses:**

1.	The presentation of this paper may be improved for coherence. For example, in Sec. 3.3, the GCB module is designed/modified based on Hyena, the key component for sub-quadratic complexity. The authors may want to include a short description of it in the main context rather than the appendix. Otherwise, it may introduce difficulties in comprehension. In addition, the proposition in the main context assists in analyzing the complexity, which is presented in the appendix. It seems like they can be excluded from the main context.
2.	The motivation is to make the model parameters sub-quadratic to the number of nodes. While theoretical analysis is conducted, I would like to see empirical results (e.g., training time) in GECO’s training efficiency compared with other baselines.

**Questions:**

1.	In the Graph-to-sequence conversion part, the authors state “time-correlated sequences, aligning node IDs with time (t)”. Where does the ‘time’ come from? What does it mean?
2.	The LCB module conducts neighborhood information propagation for each node. It directly utilizes the connectivity information via adjacency matrix. In the meantime, GECO is implicitly learning this ‘connectivity’ via the convolutional filters. Is there any information overlap here?

---

> ### Author Response · Authors · 2023-11-17
>
> Thank you for a thorough and insightful review.  Below we address the raised questions and concerns.
>
>
> > @W1 "The presentation of this paper may be improved for coherence. For example, in Sec. 3.3, the GCB module is designed/modified based on Hyena, the key component for sub-quadratic complexity. The authors may want to include a short description of it in the main context rather than the appendix. Otherwise, it may introduce difficulties in comprehension. In addition, the proposition in the main context assists in analyzing the complexity, which is presented in the appendix. It seems like they can be excluded from the main context."
>
> Thanks for your suggestion. We have reorganized the text as suggested in order to improve clarity and comprehension. Please see the revised version of the manuscript.
>
> > @W2 "The motivation is to make the model parameters sub-quadratic to the number of nodes. While theoretical analysis is conducted, I would like to see empirical results (e.g., training time) in GECO’s training efficiency compared with other baselines."
>
> Thank you for your suggestion. Please see the general rebuttal above.
>
> > @Q1 "In the Graph-to-sequence conversion part, the authors state “time-correlated sequences, aligning node IDs with time (t)”. Where does the ‘time’ come from? What does it mean?"
>
> The original operator was initially designed for sequence models, where the input has a specific order correlated with position t. In this context, two consecutive tokens represents time points t and (t + 1). We aim to clarify how this concept applies to our scenario. As we explained in Section 3.4, our approach involves initially permuting the graph with a specific ordering. Then, we treat these ordered vertices as a sequence.
>
> > @Q2 "The LCB module conducts neighborhood information propagation for each node. It directly utilizes the connectivity information via adjacency matrix. In the meantime, GECO is implicitly learning this ‘connectivity’ via the convolutional filters. Is there any information overlap here?"
>
> Yes, there is a slight information overlap. LCB block tells us where to pay attention using adjacency matrix, however if adjacency lacks information due to missing/spurious edges, GCB compliments it by enabling information flow from the rest of the vertices.
>
> The ablation study at Table 5 can be seen as an empirical support for our motivation to use LCB and GCB together. For clarity, we will use the relative rows of Table 5 below:
>
> | Model         | PascalVOC-SP   | Peptides-func   | Peptides-struct   |
> | ------------- | -------------- | --------------- | ----------------- |
> |               | F1 score ↑     | AP ↑            | MAE ↓             |
> | GECO (Conv-1)| 0.2752         | 0.6589          | 0.2587            |
> | GECO (LCB)   | **0.3220**         | **0.6876**          | **0.2454**            |
>
> Above, Conv-1 refers to an alternative approach in GECO where LCB is substituted with a 1D-Convolution having a filter size of 1, essentially replacing LCB with an identity function.  This scenario represents the case where we do not utilize the adjacency matrix and explicitly learn connectivity information using only GCB. Notably, when LCB is employed, we observe significant improvements in predictive performance across all datasets.

---

> ### Author Response · Authors · 2023-11-22
> **Kindly Reminder**
>
> Dear Reviewer vPHt,
>
> We wanted to follow up to see if the response and revisions have addressed your concerns. We would be happy to provide further clarifications and revisions if you have any more questions. If not, we would greatly appreciate it if you would reevaluate our paper. Thank you again for your reviews, which have helped improve our paper!

---

### Official Review · Reviewer_XpCe · 2023-11-02

**Soundness:** 2 fair
**Presentation:** 2 fair
**Contribution:** 1 poor
**Rating:** 3
**Confidence:** 5

**Summary:**

This article proposes a new operator-GECO to replace the graph transformer to solve the computational complexity problem of MHA (multi-head attention) on large-scale graphs. GECO introduces the Hyena architecture into graph convolution calculations, using a combination of long convolutions and gating to compute local and global context. Subsequent experiments have proven that GECO can ensure accuracy while reducing time complexity, on large-scale and small-scale data sets. The main contributions of the article are 1. There is no trade-off between quality and scalability while ensuring both; 2. It confirms that the Hyena architecture can replace MHA in graph neural networks, and global context can improve the performance of GNN.

**Strengths:**

A new operator-GECO to replace the graph transformer to solve the computational complexity problem of MHA (multi-head attention) on large-scale graphs.

**Weaknesses:**

- The technical contribution is limited. According to  this survey [2], the proposed LCB module can be treated as the GNN-as-Auxiliary-Modules in the Alternatively form (Figure 1 in [2]). Additionally, the writing of this paper seems rushed. Many details are missing and hard to understand. For example, in Algorithm 1, line 3, what is $V_t \leftarrow (P)_t FFTConv(F_i, V)_t$. Actually, I found more details of this algorithm in Algorithm 3,  page 8, [link](https://arxiv.org/pdf/2302.10866.pdf)[1]. The forward pass of GCB Operator is nearly identical to Hyena, which is not new.

- Using positional  embedding to encode the graph structural information is not new.

- The paper claims that the proposed model is "fast", and provides detailed time complexity analysis. Unfortunately, from the theoretical perspective, GECO has the same level complexity as Message-passing GNN $O(NlogN+M)$ and it can only surpass vallia transformer when $M<<N^2$. Additionally, no experiments regarding the running time efficiency are presented.

- It's necessary make more comparisons with more baselines of Graph Transformer. Please refer to [2] for more baselines.

[1] Hyena Hierarchy: Towards Larger Convolutional Language Models
[2] Transformer for Graphs: An Overview from Architecture Perspective

**Questions:**

See weakness.

**Details Of Ethics Concerns:**

None.

---

> ### Author Response · Authors · 2023-11-17
>
> We thank the reviewer for an insightful review. Below we address each of the individual concerns.
>
> > @W1 "The technical contribution is limited. According to this survey [2], the proposed LCB module can be treated as the GNN-as-Auxiliary-Modules in the Alternatively form (Figure 1 in [2])."
>
> Thanks for pointing out the related work which we included in the revised manuscripts.
>
> Approaches combining of-the-shelf GNNs and attention mechanisms were already mentioned in our submission. In GraphTrans [3] (Before Trans type in [2]), GNN and Transformer blocks are applied sequentially, while GraphGPS [4] (Parallel type in [2]) employs them in parallel at each layer. It is worth noting that these models straightforwardly combine pre-existing GNN and Transformer models, resulting in separate parameters and intermediate non-linearities for each module, with skip connections applied independently.
>
> **GECO does not precisely align with the taxonomy defined in [2].** In GECO, we did not just use LCB as an auxiliary module to Transformer. Instead, we designed a new compact layer comprising local and global blocks. We eliminated the intermediate non-linearities and parameters to reduce the overall number of parameters, simplifying the model. We applied skip connections to the entire GECO layer as a whole, rather than separately. These deliberate design choices distinguish GECO from the use of off-the-shelf methods.
>
> Mesh Graphormer [5], the only related work categorized under 'Alternatively form' in [2], similarly diverges from GECO regarding intermediate non-linearities, parameters, and skip connections, as discussed above. It also differs in the order of local and global modules, with Mesh Graphormer placing attention before the GNN block, whereas in GECO, local follows global.
>
>
> > @W2 "Additionally, the writing of this paper seems rushed. Many details are missing and hard to understand. For example, in Algorithm 1, line 3, what is V_t \leftarrow (P)_t FFTConv(F_i, V)_t."
>
> In Section 3.3's last paragraph, we detail Algorithm 1's logic and aim to enhance clarity with pseudocode comments. The GCB operator conducts global convolutions sequentially for projections and filters, gating output with subsequent results. P_i is for projections, F_i for filters, and V for values in pseudocode comments. FFTConv represents FFT Convolutions. We added more clarifications to the pseudocode in the revised manuscript.
>
> We have put genuine effort into creating a comprehensive manuscript that includes details on related work, background, pseudocode, implementation, and dataset information, with appropriate citations for further details. If there are particular details you would like us to provide, please let us know.

---

> ### Author Response · Authors · 2023-11-17
>
> > @W2 "Actually, I found more details of this algorithm in Algorithm 3, page 8, link[1]. The forward pass of GCB Operator is nearly identical to Hyena, which is not new."
>
> We list the two design innovations of our work in our previous response to @W1 and also in the general rebuttal.
>
> The last paragraph of Section 3.3 already emphasizes that we adapt and modify the notation used in Hyena for Algorithm 1 with proper citation. Our novelty for GCB is centered on the design of the convolution filters and their utilization for graph domain. We list 4 main design points for GCB: (1) Graph-to-Sequence conversion, (2) Bi-directional information flow, (3) Graph-aware context, and (4) Window of the global convolution. On the other hand, Algorithm 1 (or Alg. 3 [1]) is a generic pseudocode for models following the Hyena architecture. Below, we highlight some differences in the global convolution pseudocodes using comments labeled as "Difference:"
>
> ```
> Algorithm 1 [2]: Projection
> Input: - Input sequence u in R^(L x D)
>
> 1. In parallel across L: \hat{z} = Linear(u), Linear: R^D -> R^((N+1)D)
> 2. In parallel across D: z = DepthwiseConv1d(h, \hat{z}) # ----> Difference 1: Removed 1D conv as position under given node ordering, does not imply promixity. Refer to Section 3.3.
> 3. Reshape and split z into x^1, ..., x^N, v. Dimensions of one element are x^n in R^(D x L)
> Return x^1, ..., x^N, v, x^n
> ```
>
>
> ```
> Algorithm 2 [1]: HyenaFilter
> Input: - Sequence length L, positional embedding dimension D_e #
> 1. t = PositionalEncoding(L), t in R^(L x D_e)
> 2. In parallel across N, L: \hat{h} = FFN(t), FFN: R^D_e -> R^(N D), \hat{h} in R^(L x N D)
> 3. Reshape to \hat{h} in R^(N x D x L)
> 4. h = \hat{h} * Window(t), h in R^(N x D x L) #  ----> Difference 2: Removed decay, we need to treat all nodes equally, regardless of their distance under given ordering. Refer to Section 3.3.
> 5. Split h into h^1, ..., h^N
> Return h^1, ..., h^N
> ```
>
>
> ```
> Algorithm 3 [1]: Hyena Operator
> Input: Input sequence u in R^(L x D), order N, model width D, sequence length L, positional embedding dimension D_e
> #  ----> Difference 3: Set L = N, to ensure global convolution encapsulates all nodes. Refer to Section 3.3.
>
> 1. x^1, ..., x^N, v = Projection(u)
> 2. h^1, ..., h^N = HyenaFilter(L, D_e)
>
> For n = 1,...,N
>     3. In parallel across D: v_t <- x^n_t * FFTConv(h^n, v)_t  ----> Difference 4: # Removed causality to enable bi-directional information blow between vertices. Refer to Section 3.3.
> Return y = v
> ```
>
> ```
> Algorithm 5 (Ours): Forward GECO Operator
> Input: - Adjacency matrix Adj in R^(N x N) - Node embeddings X in R^(N x d)
>
> 1. X = BN( Propagate(X, Adj) ) #  ----> Difference 5: Incorporated LCB to capture local dependencies. Refer to Section 3.3.
> #  ----> Difference 6: We apply normalization after local block
> 2. X = GCB(X, Adj)
> Return X
> ```
>
> To state that our method is "identical" would suggest that the off-the-shelf Hyena is on par with its graph adaptation counterpart, GCB. However, this is not the case. To reinforce our argument, we conducted the ablation study below. In this study, we employed off-the-shelf Hyena, with one modification – setting the window size as the entire graph (Difference 3). The results demonstrate that GECO consistently outperforms off-the-shelf Hyena, which exhibits significant quality declines across all datasets. We added these results to the revised manuscript as well.
>
>
> | Model         | Flickr  | Yelp   | Reddit | Arxiv   |
> |---------------|:------------:|:--------:|:--------:|:----------------:|
> | Off-the-shelf Hyena [1] |   46.97 ± 0.08  |  50.08 ± 0.31 | 69.24 ± 0.54  | 56.04 ± 0.61|
> | GECO   |  **55.55 ± 0.25**          | **63.18 ± 0.59**     | **96.65 ± 0.05**   | **73.10 ± 0.24**           |
> | Relative Improvement (%)   | 18.24% | 26.11% | 39.57% | 30.39% |
>
>
> Please further refer to ablation study at Table 5, showing that omitting some elements of design choice 2 (Difference 1 & 6), "Graph-aware context," has also significant effects, reducing quality by up to 55%.
>
> > @W3 "Using positional embedding to encode the graph structural information is not new."
>
> We agree. This is not new, consequently we do not claim any novelty with the usage of positional/structural encodings within our framework. In Section 3.1 we note that we follow the foundational work laid out by the literature. However, we need to provide the reader necessary information on how to train our model end-to-end including the usage of positional/structural encodings.

---

> ### Author Response · Authors · 2023-11-17
>
> > @W4 "The paper claims that the proposed model is "fast", and provides detailed time complexity analysis.  Unfortunately, from the theoretical perspective, GECO has the same level complexity as Message-passing GNN  and it can only surpass vallia transformer when M << N^2"
>
> **M << N^2 always holds for real-world datasets.** This property is the main reason why we need sparse computations to train GNNs. To put in more concrete form, we provide a table where we explicitly calculate M/N^2 for each dataset we use. Specifically, large node prediction, the graphs are even sparser. We also added these columns to the revised version of the manuscript.
>
> | Dataset       |      N       |    M     |    M / N^2       |
> |---------------|:------------:|:--------:|:----------------:|
> | PCQM-Contact  |     30.1     |   61.0   | 6.79 x 10^-2      |
> | Peptides-func |    150.9     |  307.3   | 1.36 x 10^-2      |
> | Peptides-struct|   150.9     |  307.3   | 1.36 x 10^-2      |
> | COCO-SP       |    476.9     | 2,693.7  | 1.20 x 10^-2      |
> | PascalVOC-SP  |    479.4     | 2,710.5  | 1.20 x 10^-2      |
> | PCQM4Mv2      |     14.1     |   14.6   | 7.25 x 10^-2      |
> | Molhiv        |     25.5     |   27.5   | 4.29 x 10^-2      |
> | Molpcba       |     26.0     |   28.1   | 4.13 x 10^-2      |
> | Code2         |    125.2     |  124.2   | 1.59 x 10^-2      |
> | PPA           |    243.4     | 2,266.1  | 3.25 x 10^-2      |
> | Flickr        |   89,250     | 899,756  | 1.12 x 10^-4      |
> | ogbn-arxiv    |  169,343     | 1,166,243| 3.97 x 10^-5      |
> | Reddit        |  232,965     |114,615,892| 1.95 x 10^-6     |
> | Yelp          |  716,847     |13,954,819| 2.26 x 10^-6      |
>
> Additionally, while our approach entails higher complexity than MPNN, it exhibits near-linear complexity which is a significant advancement compared to graph transformers with dense attention, all the while having a robust predictive quality, as demonstrated through Table 1-4.
>
> > @W4 "Additionally, no experiments regarding the running time efficiency are presented."
>
> Thank you for your suggestion. Please see the general rebuttal above.

---

> ### Author Response · Authors · 2023-11-17
>
> > @W5 "It's necessary make more comparisons with more baselines of Graph Transformer. Please refer to [2] for more baselines."
>
> Thanks for pointing out the related work. Below, we present a comparison with [2], selecting the best variant for each model/dataset configuration. Our comparison covers common datasets used by both works, encompassing all datasets in [2], except for two.
>
> We highlight that GECO consistently outperforms the models in [2] across all datasets. Specifically, on arxiv and molphcha, GECO achieves significant relative improvements of up to 28.11% and 11.23%, respectively. We also note that, some approaches discussed in [2] and reported below such as positional encoding (PE) and improved attention matrices from graphs (AT) are orthogonal approaches to our work. We have also incorporated these comparisons in the revised version of the manuscript.
>
> |      |                   | molhiv | molpcba | Flickr | ogbn-arxiv |
> |------|-------------------|--------|---------|--------|------------|
> |      |                   |  ROC-AUC↑|   AP↑   |   Acc↑ |   Acc↑     |
> | TF   | vanilla           |  0.7466 | 0.1676  | 0.5279 | 0.5598     |
> | GA   | before            | 0.7339 | 0.2269  | 0.5369 | 0.5614     |
> |      | alter             | 0.7433 | 0.2474  | 0.5374 | 0.5599     |
> |      | parallel          | 0.7750 | 0.2444  | 0.5379 | 0.5647     |
> | PE   | degree            |  0.7506 | 0.1672  | 0.5291 | 0.5618     |
> |      | eig               | 0.7407 | 0.2194  | 0.5278 | 0.5658     |
> |      | svd               | 0.7350 | 0.1767  | 0.5317 | 0.5706     |
> | AT   | SPB               |  0.7589 | 0.2621  | 0.5368 | 0.5605     |
> |      | PMA               |  0.7314 | 0.2518  | 0.5288 | 0.5571     |
> |      | Mask-1            | 0.7960 | 0.2662  | 0.5300 | 0.5598     |
> |      | Mask-n            |  0.7423 | 0.2619  | 0.5359 | 0.5603     |
> | GECO  (Ours) |   | **0.7980 ± 0.0200**       | **0.2961 ± 0.0008**               | **0.5555 ± 0.0025**           | **0.7310 ± 0.0024**   |
>
>
> Additionally, directly quoting from [2] is two of the three feature directions highlighted in the survey, which perfectly align with our motivations:
>
> 1. "**New paradigm of incorporating the graph and the Transformer**: Most studies treat the graphs as strong prior Transformer model. There is a great interest to develop the new paradigm that not just takes graphs as a prior, but also better reflects the properties of graphs." [2]
> 2. "**Extending to large-scale graphs.** Most existing methods are designed for small graphs, which might be computationally infeasible for large graphs. As illustrated in our experiments, directly applying them to the sampled subgraphs would impair performance. Therefore, designing salable Graph-Transformer architecture is essential." [2]
>
> Moreover, Table 1 in [2] lists various models, many of which our evaluation already covers, including well-known methods like Graphormer, GraphiT, EGT, SAN, and GraphTrans. Our submission also included additional models not in the list, such as GraphGPS and Exphormer. In our revised version, we incorporated more baselines from [6], as suggested by R-vPHt. If there is a particular comparison you would like us to provide, please let us know.
>
> [1] Hyena Hierarchy: Towards Larger Convolutional Language Models, ICML'2023
>
> [2] Transformer for Graphs: An Overview from Architecture Perspective, Arxiv'2022
>
> [3] Do transformers really perform badly for graph representation?, NeurIPS'21
>
> [4] GraphGPS: General Powerful Scalable Graph Transformers, NeurIPS'22
>
> [5] Mesh Graphormer, ICCV'2021
>
> [6] Hierarchical Transformer for Scalable Graph Learning, IJCAI'23

---

> ### Author Response · Authors · 2023-11-22
> **Kindly Reminder**
>
> Dear Reviewer XpCe,
>
> We wanted to follow up to see if the response and revisions have addressed your concerns. We would be happy to provide further clarifications and revisions if you have any more questions. If not, we would greatly appreciate it if you would reevaluate our paper. Thank you again for your reviews, which have helped improve our paper!

---

### Author Response · Authors · 2023-11-17

We appreciate the reviewers for their time and thoughtful feedback. It is encouraging that our motivation was found to be good, significant, and meaningful (R-vPHt, R-Utmf), with our improvements noted as very smart (R-Utmf) and promising (R-vPHt). We are glad that our evaluation was seen as extensive (R-vPHt), its content as rich (R-Utmf), covering both small and large datasets (R-XpCe) and complemented by an appendix for clear understanding of the paper (R-Utmf). We are also pleased that the main contributions of our work were acknowledged: (1) With GECO, there is no trade-off between quality and scalability, ensuring both. (2) Global convolutions can replace the dense attention mechanism in graph transformer models (R-vPHt, R-Utmf).

However, although our complexity analysis has been noticed, one primary concern raised is the lack of empirical runtime/scalability evaluation among all reviewers. We understand this concern and, will begin our rebuttal by providing clarifications and addressing it, followed by responses to individual concerns and questions in detail. Furthermore, we will outline the revisions made in our manuscript.

---

> ### Author Response · Authors · 2023-11-17
> **Clarifications, and Scalability and Runtime Evaluations**
>
> **Motivation:** As noted by R-vPHt, our goal is to reduce the model complexity to sub-quadratic in relation to the number of nodes while achieving competitive or superior predictive quality w.r.t self attention mechanism. Proposed model, GECO, has a complexity of O(N log N + M), while traditional dense GTs exhibit O(N^2) or O(N^2 + M) complexity. Our innovations are:
>
> 1. **Designing a compact layer consisting of local context and global context blocks (GECO).** Unlike other approaches that straightforwardly combine off-the-shelf GNNs and Transformers, we design our layer carefully, by removing non-lineraties and excess parameters to simplify the model. Refer to our discussion at R-XpCe @W1 for more details.
>
> 2. **Designing a global convolutional model for graphs.**  We demonstrate a promising approach to replace GT's self-attention with global convolution to effectively implement global convolution-based architectures in graph domains, by adapting Hyena[1] to graphs. This adaptation serves as the global context block within our novel GECO layer. Please refer to our discussions at R-XpCe and R-Utmf for more information.
>
> **Dataset Clarification.** The average number of nodes Table 1, 2 and 3 range from 30 to 480. These datasets have many data points, however each data point is very small. Consequently, scalability is not a significant concern for them since the number of average nodes determines computational load. As an evidence, even the most computation-intensive models, such as Graphormer [2] or GraphGPS [3], can train on these datasets using Nvidia-V100 (32GB) or Nvidia-A100 (40GB) GPUs. We included these evaluations to demonstrate GECO's competitive predictive performance compared to GT baselines. Specifically, GECO outperforms GraphGPS in 8 out of 10 cases on these small datasets.
>
> **Empirical Verification for Scalability:** To support our theoretically outlined scalability claim, in Table 4, we scale up to large node prediction datasets.  As noted in our related work and literature recommended by the reviewers, traditional graph transformers with dense attention mechanisms fail to scale to any of these datasets [2, 3, 4, 5]. In contrast, GECO scales to these datasets, consistently outperforming many GNN and GT baselines.
>
> **Empirical Verification for Runtime.** However, we acknowledge concerns regarding absence of empirical runtime evaluation, particularly given that our paper is titled "Fast", which can be ambiguous. To rectify this concern, we present our experiments and findings below.
>
> *Experimental Setting.* We have generated random graphs using Erdos-Renyi model. We increased the number of nodes from 512 to 2 million by doubling the number of nodes at consecutive points. We set the sparsity factor of each graph to $10/N$, aligning the sparsity of the graph with that of large node prediction datasets in Table 4. For relative speedup figure and details refer to Section 4.3 and Appendix E.1 in revised paper.
>
> The results highlight significant runtime improvements, with GECO reaching a 169x speedup on a graph with 2M nodes, confirming its efficiency for larger-scale applications. This is anticipated due to GECO's complexity of O(Nlog N + M), while attention's complexity is O(N^2). Considering the sparsity of real-world datasets, N becomes a dominant factor, leading to a speedup characterized by O(N / log N).
>
> On the other hand, for smaller scales, the choice between the two could be influenced by factors beyond just performance. As discussed throughout our work, the scalability gains are not expected for small graphs. This is mostly due to low hardware utilization incurred by available FFT implementations. However, GECO still remains a promising approach due to its high prediction quality, as we demonstrated in Section 4.1. On larger graphs, GECO exhibits significant scalability, as demonstrated in Section 4.2. It consistently outperforms dense GTs on all large datasets and remains superior or competitive when compared to the orthogonal approaches.
>
>
> |   N    | GECO (ms) | FlashAttention (ms) | Relative Speedup |
> |-------|-----------|---------------------|------------------|
> |  512  |   1.88    |        0.27         |       0.14       |
> | 1,024 |   2.13    |        0.32         |       0.15       |
> | 2,048 |   2.11    |        0.31         |       0.15       |
> | 4,096 |   2.42    |        0.32         |       0.13       |
> | 8,192 |   2.12    |        0.51         |       0.24       |
> |16,384 |   2.13    |        1.84         |       0.86       |
> |32,768 |   2.63    |        6.92         |       2.63       |
> |65,536 |   3.73    |       28.74         |       7.70       |
> |131,072|   6.21    |      115.23         |      18.56       |
> |262,144|  15.74    |      458.64         |      29.14       |
> |524,288|  41.72    |     1830.29         |      43.87       |
> |1,048,576| 83.90  |     7317.04         |      87.21       |
> |2,097,152|173.15 |    29305.77         |     169.25       |

---

> ### Author Response · Authors · 2023-11-17
> **Manuscript Revisions**
>
> We have,
>
> - Included a study and discussion on runtime scaling as requested by all reviewers in Section 4.3 and Appendix E.1.
> - Included ablation study and discussing comparing GECO with off-the-shelf Hyena, demonstrating effectiveness of our method in Section 4.3. Also changed the orientation of Table 6.
> - Included comparison and discussion on HSGT [4] and its baselines in Section 4.2.
> - Included a comparison and discussion in relation to the survey conducted in [6] in the Appendix.
> - Generalized Section 3.5 as a discussion comparing various hybrid approaches with GECO's architectural design.
> - Clarified contributions in Section 1, budget constraints for Table 1 in the Appendix, graph sparsity properties in Tables 7 & 8, and Algorithm 3's notation. We also moved the Hyena description into Section 2.3.
>
>  Due to space constraints, we have moved/added/excluded various text in order to accommodate the above revisions. Unfortunately, we could not include all additional studies and comparisons in the main text; some are placed in the appendix.
>
> [1] Hyena Hierarchy: Towards Larger Convolutional Language Models, ICML'2023
>
> [2] Do Transformers Really Perform Bad for Graph Representation?, NeurIPS'21
>
> [3] GraphGPS: General Powerful Scalable Graph Transformers, NeurIPS'22
>
> [4] Hierarchical Transformer for Scalable Graph Learning, IJCAI'23
>
> [5] EXPHORMER: Sparse Transformers for Graphs, ICML'23
>
> [6] Transformer for Graphs: An Overview from Architecture Perspective, Arxiv'2022
>
> [7] FlashAttention: Fast and Memory-Efficient Exact Attention with IO-Awareness, NeurIPS'22

---

### Meta-Review · Area_Chair_ydPQ · 2023-12-08

**Metareview:**

The main contribution of GECO is a faster graph transformer inspired by the Hyena LLM, which replaces the transformer global attention with a sequence model (Hyena uses a k-order Markovian convolutional filter). This makes the previously permutation-equivariant transformer architecture permutation-sensitive (i.e., closer to a sequence model, not relying as much on positional encodings). The training randomly permutes the features and adjacency matrix in the hope that the model will be less permutation-sensitive by doing so. I can see the argument of the approach being theoretically sound as (Murphy et al., ICML 2019) had shown that this type training is an ELBO for an equivalent permutation-invariant/equivariant model from a sequence model via symmetrization.

The traditional spectral positional embedding for transformers in GECO is made further permutation-sensitive by a model that is itself permutation-sensitive. We know that making GNNs permutation-sensitive by adding permutation-sensitive features increase their expresiveness (Murphy et al., ICML 2019; Vignac et al., NeurIPS 2020; Sato et al. SDM 2021). One may then wonder if there could be a similar property for graph transformers. The authors did not specifically frame their gains with respect to this change. The work seems to be more focused on scalability.

Overall the reviewers were not supportive of the work, minimally there were no champions of the work. Some of the concerns were well addressed by the authors in the rebuttal (e.g., speed comparisons, whether the sparsity assumption is justified). The question whether GECO has enough novelty w.r.t. Hyena is somewhat subjective but raised by multiple reviewers. I think if the authors had a better theoretical justification for their approach, it would have been harder to argue for a lack of novelty. Why should GECO outperform other transformer approaches? E.g., show a simple family of graphs where GECO is supposed to outperform other transformer methods that supports the claims why having a more permutation sensitive model improves things (both in speed and accuracy). Graphs are not natural language text whose data topology is unknown: If a graph model works better there is likely a structural reason to why.

**Justification For Why Not Higher Score:**

If the authors had given better justification for then algorithm choices in GECO, I feel the paper should have been accepted. As it stands, it feels a bit like "Hyena" is a new LLM method, let's try to port it to graphs. It is faster but it is also more permutation sensitive. There should be a better justification for its use.

**Justification For Why Not Lower Score:**

N/A

---

### Decision · Program_Chairs · 2024-01-16

Reject